# How Transformers Learn Regular Language Recognition: A Theoretical Study on Training Dynamics and Implicit Bias

Ruiquan Huang [1]   Yingbin Liang [2]   Jing Yang [3]

## Abstract

Language recognition tasks are fundamental in natural language processing (NLP) and have been widely used to benchmark the performance of large language models (LLMs). These tasks also play a crucial role in explaining the working mechanisms of transformers. In this work, we focus on two representative tasks in the category of regular language recognition, known as 'even pairs' and 'parity check', the aim of which is to determine whether the occurrences of certain subsequences in a given sequence are even. Our goal is to explore how a one-layer transformer, consisting of an attention layer followed by a linear layer, learns to solve these tasks by theoretically analyzing its training dynamics under gradient descent. While even pairs can be solved directly by a one-layer transformer, parity check need to be solved by integrating Chain-of-Thought (CoT), either into the inference stage of a transformer well-trained for the even pairs task, or into the training of a one-layer transformer. For both problems, our analysis shows that the joint training of attention and linear layers exhibits two distinct phases. In the first phase, the attention layer grows rapidly, mapping data sequences into separable vectors. In the second phase, the attention layer becomes stable, while the linear layer grows logarithmically and approaches in direction to a max-margin hyperplane that correctly separates the attention layer outputs into positive and negative samples, and the loss decreases at a rate of $O(1/t)$. Our experiments validate those theoretical results.

[1]Department of Electrical Engineering, The Pennsylvania State University, State College, PA 16801, USA [2]Department of Electrical and Computer Engineering, The Ohio State University, Columbus, OH 43210, USA [3]Department of Electrical and Computer Engineering, University of Virginia, Charlottesville, VA 22904, USA. Correspondence to: Jing Yang <yangjing@virginia.edu>.

*Proceedings of the 42nd International Conference on Machine Learning*, Vancouver, Canada. PMLR 267, 2025. Copyright 2025 by the author(s).

## 1. Introduction

Transformers (Vaswani et al., 2017) have become foundational in modern machine learning, revolutionizing natural language processing (NLP) tasks such as language modeling (Devlin et al., 2018), translation (Wang et al., 2019), and text generation (Radford et al., 2019). Among these, **language recognition tasks** are fundamental to NLP and are widely used to benchmark the empirical performance of large language models (LLMs) (Bhattamishra et al., 2020; Deletang et al., 2023). Beyond their practical applications, these tasks hold significant potential for uncovering the underlying working mechanism of transformers. A growing body of research has explored the expressiveness and learnability of transformers in these settings (Strobl et al., 2024; Hahn & Rofin, 2024; Chiang & Cholak, 2022; Merrill & Sabharwal, 2023; Hahn, 2020). Despite this, there has been little effort to understand transformers' training dynamics in language recognition tasks.

In this work, we take the first step towards bridging this gap by focusing on two fundamental pattern recognition tasks in formal language recognition, known as 'even pairs' and 'parity check' problems, and explore how transformers can be trained to learn these tasks from a theoretical perspective. Specifically, the objective of the 'even pairs' problem is to determine whether the total number of specific subsequences in a binary sequence is even, and the objective of the 'parity check' problem is to determine whether the total occurrence of a single pattern is even. These tasks are particularly compelling for studying transformers because they require the model to recognize parity constraints and capture global dependencies across long sequences, which are essential for real-world applications such as syntax parsing and error detection in communication systems.

For the two problems of our interest, the even pairs problem has not been studied before theoretically. The parity check problem has recently been studied in Kim & Suzuki (2024b); Wen et al. (2024), which characterized the training dynamics of CoT for learning parity. However, Kim & Suzuki (2024b) analyzed the training of an attention layer only, leaving more general characterization of joint training of feed-forward and attention layers yet to be studied. Wen et al. (2024) analyzed three iteration steps in training

without establishing the convergence of the entire training process. Our goal is to develop a more general training dynamics characterization, including joint training of the attention and linear feed-forward layers, the convergence rate of the loss functions, and the implicit bias of the training parameters. Furthermore, while both the 'even pairs' and 'parity check' problems are classification tasks, existing theoretical studies on transformers in classification settings (Li et al., 2023; Tarzanagh et al., 2023b;a; Vasudeva et al., 2024; Deora et al., 2023; Yang et al., 2024a; Magen et al., 2024; Jiang et al., 2024; Sakamoto & Sato, 2024) have primarily focused on cases where class distinctions are based on *identifiable* features. In contrast, these language recognition tasks will pose unique challenges, which require the transformer to leverage its attention mechanism to uncover intricate dependencies inherent in data sequences. By exploring these tasks, our work will offer new insights into the fundamental mechanism of transformers.

In this work, we investigate how a one-layer transformer, consisting of an attention layer followed by a linear layer, learns to perform the 'even pairs' and 'parity check' tasks. We will theoretically analyze the model dynamics during the training process of gradient descent, and examine how transformer parameters will be guided to converge to a solution with implicit bias. Here, we will jointly analyze the training process of the attention layer and linear layer, which will be significantly different from most existing analysis of the training dynamics of transformers for classification problems, where joint training is not studied (Huang et al., 2024a; Tarzanagh et al., 2023b; Kim & Suzuki, 2024b; Li et al., 2024b).

Our major contributions are four-fold:

First, for the even pairs problem, we identify two distinct learning phases. In Phase 1, both linear and attention layers grow rapidly, inducing separable outputs of the attention layer. In Phase 2, the attention layer remains almost unchanged, while the dynamics of the linear layer is governed by an implicit bias, which converges in direction to the max-margin hyperplane that correctly separates the attention layer's outputs into positive and negative samples. We also show that the loss function decays to the global minimum sublinearly in time. To the best of our knowledge, this is the first theoretical study on the training dynamics of transformers for the even pairs problem.

Second, we innovatively leverage the insights from the even pairs problem and Chain-of-Thought (CoT) to solve the parity check problem through two different approaches. In the first approach, we introduce truncated CoT into the *inference* stage of a trained transformer. We demonstrate that a transformer, well-trained on the even pairs problem but without CoT training, can successfully solve the parity check problem in a zero-shot manner (without any additional train-

ing) using truncated CoT inference. Such a surprising result is based on the intricate connection between the even pairs problem and the parity check problem. For the second approach, it trains a one-layer transformer with CoT under teacher forcing, where we further include the training loss of even pairs to stabilize the training process. We show that with a two-phase training process, similarly to that of the even pairs problem, gradient descent provably renders a one-layer transformer that can solve parity check via CoT.

Third, we introduce a novel analytical technique for studying joint training of attention and linear layers. Specifically, we employ higher-order Taylor expansions to precisely analyze the coupling between gradients of two layers and its impact on parameter updates in Phase 1. Then we incorporate implicit bias principles to further characterize the training dynamics in Phase 2. Since all parameters are actively updated, we must bound the perturbations in the attention layer and analyze their effects on the linear layer. This is achieved by carefully designing the scaling factor in the attention mechanism, which not only stabilizes training but also underscores its critical role in transformer architectures.

Finally, we conduct experiments to validate our theoretical findings, demonstrating consistent parameter growth, alignment behavior, and loss convergence.

## 2. Related Work

Due to the recent extensive theoretical studies of transformers from various perspectives, the following summary will mainly focus on the training dynamics characterization for transformers, which is highly relevant to this paper.

**Learning regular language recognition problems via transformers.** Regular language recognition tasks are fundamental to NLP and are widely used to benchmark the empirical performance of large language models (LLMs) (Bhattamishra et al., 2020; Deletang et al., 2023). Theoretical understanding of transformers for solving these tasks are mainly focusing on the expressiveness and learnability Strobl et al. (2024); Hahn & Rofin (2024); Chiang & Cholak (2022); Merrill & Sabharwal (2023); Hahn (2020). Among these studies, several negative results highlight the limitations of transformers in learning problems such as parity checking. Notably, Merrill & Sabharwal (2023) demonstrated that chain-of-thought (CoT) reasoning significantly enhances the expressive power of transformers. We refer readers to the comprehensive survey by Strobl et al. (2024) for a detailed discussion on the expressiveness and learnability of transformers. Regarding the study on the training transformers, Kim & Suzuki (2024b); Wen et al. (2024) showed that it is impossible to successfully learn the parity check problem by applying transformer once. They further developed CoT method and showed that attention model

with CoT can be trained to provably learn parity check. Differently from Kim & Suzuki (2024b), our work analyzed the softmax attention model jointly trained with a linear layer with CoT for learning parity check. Our CoT training design is also different from that in Kim & Suzuki (2024b). Further, Wen et al. (2024) studied the sample complexity of training a one-layer transformer for learning complex parity problems. They analyzed three steps in the training procedure without establishing the convergence of the entire training process, which is one of our focuses here.

**Training dynamics of transformers with CoT.** CoT is a powerful technique that enables transformers to solve more complex tasks by breaking down problem-solving into intermediate reasoning steps (Wei et al., 2022; Kojima et al., 2022). Recently, training dynamics of transformers with CoT has been studied in Kim & Suzuki (2024b); Wen et al. (2024) for the parity problems (as discussed above) and in Li et al. (2024a) for in-context supervised learning.

**Training dynamics of transformers for classification problems.** A recent active line of research has focused on studying the training dynamics of transformers for classification problems. Tarzanagh et al. (2023b;a) showed that the training dynamics of an attention layer for a classification problem is equivalent to a support vector machine problem. Further, Vasudeva et al. (2024) established the convergence rate for the training of a one-layer transformer for a classification problem. Li et al. (2023) characterized the training dynamics of vision transformers and provide a converging upper bound on the generalization error. Deora et al. (2023) studied the training and generalization error under the neural tangent kernel (NTK) regime. Yang et al. (2024a) characterized the training dynamics of gradient flow for a word co-occurrence recognition problem. Magen et al. (2024); Jiang et al. (2024); Sakamoto & Sato (2024) studied the benign overfitting of transformers in learning classification tasks. Although the problems of our interest here (i.e., even pairs and parity check) generally fall into the classification problem, these language recognition tasks pose unique challenges that require transformer to leverage its attention mechanism to uncover intricate dependencies inherent in data sequences, which have not been addressed in previous studies of the conventional classification problems.

**Training dynamics of transformers for other problems.** Due to the rapidly increasing studies in this area, we include only some example papers in each of the following topics. In order to understand the working mechanism of transformers, training dynamics has been intensively investigated for various machine learning problems, for example, in-context learning problems in Ahn et al. (2024); Mahankali et al. (2023); Zhang et al. (2023); Huang et al. (2023); Cui et al. (2024); Cheng et al. (2023); Kim & Suzuki (2024a); Nichani et al. (2024); Chen et al. (2024); Chen & Li (2024);

Yang et al. (2024b), next-token prediction (NTP) in Tian et al. (2023a;b); Li et al. (2024b); Huang et al. (2024a); Thrampoulidis (2024), unsupervised learning in Huang et al. (2024b), regression problem in Boix-Adsera et al. (2023), etc. Those studies for the classification problem and CoT training have been discussed above.

**Implicit bias.** Our analysis of the convergence guarantee develops implicit bias of gradient descent for transformers. Such characterization has been previously established in Soudry et al. (2018); Nacson et al. (2019); Ji & Telgarsky (2021); Ji et al. (2021) for gradient descent-based optimization and in Phuong & Lampert (2020); Frei et al. (2022); Kou et al. (2024) for training ReLU/Leaky-ReLU networks on orthogonal data. More studies along this line can be found in a comprehensive survey Vardi (2023). Most relevant to our study are the recent works Huang et al. (2024a) and Tarzanagh et al. (2023b;a); Sheen et al. (2024), which established implicit bias for training transformers for next-token prediction and classification problems, respectively. Differently from those work on transformers, our study here focuses on the even pairs and parity check problems, which have unique structures not captured in those work.

## 3. Problem Formulation

**Notations.** All vectors considered in this paper are column vectors. For a matrix $W$, $\|W\|$ represents its Frobenious norm. For a vector $v$, we use $[v]_i$ to denote the $i$-th coordinate of $v$. We use $\{e_i\}_{i \in [d]}$ to denote the canonical basis of $\mathbb{R}^d$. We use $\phi(v)$ to denote the softmax function, i.e., $[\phi(v)]_i = \exp(v_i)/\sum_j \exp(e_j^\top v)$, which can be applied to any vector with arbitrary dimension. The inner product $\langle A, B \rangle$ of two matrices or vectors $A, B$ equals $\mathrm{Trace}(AB^\top)$ $\mathrm{Trace}(AB^\top)$. For a set $\mathcal{X}$, we use $\mathcal{X}^n$ to denote the Cartesian product of $n$ copies of $\mathcal{X}$, and $\mathcal{X}^{\leq n} = \cup_{k=1}^n \mathcal{X}^k$.

In this work, we consider pattern recognition tasks in the context of formal language recognition, which challenges the ability of machine learning models such as transformers to recognize patterns over long sequences.

First, we introduce the general pattern recognition task in binary sequence. The set of all binary sequences is denoted by $\{\mathtt{a}, \mathtt{b}\}^{\leq L_{\max}}$, where $L_{\max}$ is the maximum length of the sequence. The pattern of interest is a set of sequences $\mathcal{P} = \{p_1, \ldots, p_n\} \subset \{\mathtt{a}, \mathtt{b}\}^{\leq L_{\max}}$. Given a sequence $X$ and a pattern $\mathcal{P}$, let $N_\mathcal{P}$ be the number of total matching subsequences in $X$, where a subsequence of $X$ matches if it equals some $p_i \in \mathcal{P}$. The pattern recognition task is to determine whether $N_\mathcal{P}$ satisfies a predefined condition, such as whether $N_\mathcal{P}$ is even.

In the following, we describe two representative tasks that are particularly interesting in the study of regular language recognition due to their simple formulation and inherent

learning challenges. More details about formal language recognition (which includes regular and non-regular language recognition) can be found in Deletang et al. (2023).

**Even pairs.** The pattern of interest is $\mathcal{P} = \{\text{ab}, \text{ba}\}$. For example, in the sequence $X = \text{aabba}$, there is one ab and one ba, resulting in a total of two, which is even.

**Parity check.** The pattern of interest is $\mathcal{P} = \{\text{b}\}$. In other words, this task is simply to determine whether the number of bs in a binary string $X \in \{\text{a}, \text{b}\}^L$ is even or odd.

Although *even pairs* may appear to involve more complex pattern recognition than *parity check*, it can be shown that this task is equivalent to determining whether the first and last tokens of the sequence are equal, which can be solved in $O(1)$ time (Deletang et al., 2023). However, parity check usually requires $O(L)$ time complexity given an $L$-length sequence since we need to check every token at least once.

In this work, our aim is to apply transformer models to solve these problems while leveraging these tasks to understand the underlying working mechanisms of transformer models.

**Embedding strategy.** We employ the following embedding strategy for each input binary sequence. For a token a at position $\ell$, its embedding is given by $E_\ell^{\text{a}} = e_{2\ell-1}$, and for a token b at position $\ell$, its embedding is given by $E_\ell^{\text{b}} = e_{2\ell}$. Such an embedding ensures that token embeddings are orthogonal to each other, which is a widely adopted condition for transformers.

**One-layer transformer.** We consider a one-layer transformer, denoted as $\text{T}_\theta : \mathbb{R}^{d \times L_{\max}} \to \mathbb{R}$, which takes a sequence of token embeddings as input and outputs a scaler value, and $\theta$ includes all trainable parameters of the transformer. Specifically, let $X = [x_1, \ldots, x_L]$ denote the input sequence, where $x_\ell \in \mathbb{R}^d$ for each $1 \le \ell \le L$ and $L$ is the length of the sequence. Let the value, key and query matrices be denoted by $W_v, W_k, W_q \in \mathbb{R}^{d \times d}$. Then, the attention layer is given by $W_v \sum_{\ell=1}^{L} x_\ell \varphi_\ell$, where $\varphi_\ell = [\phi(X^\top W_k W_q x_L / \lambda)]_\ell$, and $\lambda \in \mathbb{R}$ is a scaling parameter. Hence, the one-layer transformer has the form $\text{T}_\theta(X) = W_u W_v \sum_{\ell=1}^{L} x_\ell \varphi_\ell$, where $W_u \in \mathbb{R}^{1 \times d}$ denotes the linear feed-forward layer. For simplicity, we reparameterize $W_u W_v$ as $u$ and $W_k W_q$ as $W$, as commonly taken in Huang et al. (2023); Tian et al. (2023a); Li et al. (2024b). Hence, the transformer is reformulated as $\text{T}_\theta(X) = u^\top \sum_{\ell=1}^{L} x_\ell \varphi_\ell$, where $\varphi_\ell = [\phi(X^\top W x_L / \lambda)]_\ell$, and all trainable parameters are captured in $\theta = (u, W)$. We will call $W$ and $u$ respectively as attention and linear layer parameters.

When the transformer is used to for the regular language recognition tasks, for a given input $X$, it will take the sign (i.e., 1 or $-1$) of the transformer output $\text{T}_\theta(X)$ as the predicted label. For the even pairs task, a positive predicted

label 1 means the input sequence contains even pairs, and $-1$ otherwise. Similarly, for the parity check task, a positive predicted label 1 means the input sequence contains even number of the pattern of interest, and $-1$ otherwise.

**Learning objective.** We adopt the logistic loss for these binary classification tasks. We denote $I = \cup_{L=1}^{L_{\max}} I_L$ as the training dataset, where $I_L$ denotes the set of all length-$L$ sequences. An individual training data consists of a sequence $X^{(n)} = [x_1^{(n)}, \ldots, x_L^{(n)}]$ and a label $y_n \in \{1, -1\}$, where $n$ is the index of the data. With slight abuse of notation, we also use the notation $n \in I_L$ to indicate that $X^{(n)}$ has length $L$. Then, the loss function can be expressed as:

$$\mathcal{L}(u, W)$$
$$= \sum_{L=1}^{L_{\max}} \frac{1}{|I_L|} \sum_{n \in I_L} \log \left(1 + \exp\left(-y_n \text{T}_\theta(X^{(n)})\right)\right).$$

The use of $1/|I_L|$ in the loss function ensures that sequences of different lengths contribute equally to training.

Our goal is to train the transformer to minimize the loss function, i.e., to solve the problem $\min_{\theta=(u,W)} \mathcal{L}(u, W)$. We adopt a 2-phase gradient descent (GD) to minimize the loss function $\mathcal{L}(u, W)$ as follows. We adopt zero initialization, i.e., $\theta_0 = 0$. Then at early steps $t \le t_0$ (to be specified later), we update $\theta = (u, W)$ as follows:

$$u_{t+1} = u_t - \eta \nabla_u \mathcal{L}_t, \quad W_{t+1} = W_t - \eta \lambda \nabla_W \mathcal{L}_t,$$

where $\eta$ is the learning rate, and $\mathcal{L}_t$ is the abbreviation of $\mathcal{L}(u_t, W_t)$.

After step $t_0$, we update $\theta$ as follows:

$$u_{t+1} = u_t - \eta \nabla_u \mathcal{L}_t, \quad W_{t+1} = W_t - \eta \nabla_W \mathcal{L}_t$$

We remark that such a learning rate schedule can be viewed as an approximation of GD with decaying learning rate or Adam (Kingma, 2014). To be more specific, Adam updates the parameter in the form of $\theta_{t+1} = \theta_t - \eta m_t / \sqrt{v_t + \epsilon}$, where $m_t \approx g_t$, $v_t \approx g_t^2$, $\epsilon$ is a positive constant, and $g_t$ is the gradient. Thus, during the early training steps, Adam's update behaves closely to the high learning rate regime in GD. In the subsequent training steps when the gradient is relatively small, Adam behaves similar to vanilla GD due to the constant $\epsilon$ in the denominator $\sqrt{v_t + \epsilon}$.

## 4. Even Pairs Problem

In this section, we characterize the training dynamics of a one-layer transformer for learning the *even pairs* task. For simplicity, we choose $I_L = \{\text{a}, \text{b}\}^L$.

**Key challenges.** The key challenge in this analysis arises from the joint training of the linear and attention layers. The intertwined updates of these layers create a coupled stochastic process, complicating the analysis of parameter evolution.

Furthermore, since every token contributes to both positive and negative samples, this leads to gradient cancellation during analysis, making the analysis more difficult.

To address the above challenges, we employ higher-order Taylor expansions to precisely analyze the coupling between gradients of two layers and its impact on parameter updates. We next present our results for these two phases and explain the insights that these results imply. Before we proceed, we introduce the following concepts.

**Token score.** We define the token score of $E_\ell^w, w \in \{a, b\}$ as $\langle u_t, E_\ell^w \rangle$, which quantifies the alignment between the token embedding and the linear layer.

**Attention score.** For a given attention layer $W_t$, we define the (raw) attention score of $E_\ell^w$ as $\langle E_\ell^w, W_t E_L^{w'} \rangle, w, w' \in \{a, b\}$. For token $x_\ell$, its attention weight is given by $\varphi_\ell = [\phi(X^\top W_t x_L)]_\ell$, which is proportional to the exponential of its attention score. Importantly, it is the differences between the attention scores of different tokens that govern the attention weight distribution.

**Relationship with the transformer output.** Consequently, the transformer output $\mathsf{T}(X)$ for an input $X$ is a weighted average of the token scores of $x_\ell$, with the attention weights $\varphi_\ell$ serving as the weighting factors.

## 4.1. Phase 1: Rapid Growth of Token and Attention Scores

In Phase 1 of training, the linear and attention layers exhibit mutually reinforcing dynamics, with the featuring dynamics captured in the following theorem.

**Theorem 4.1** (Phase 1)**.** *Let $-w$ denote the flip of token $w \in \{a, b\}$. Choose $\lambda = \Omega(L_{\max}^2), t_0 = O(1/(\eta L_{\max}))$, and $\eta = O(\min\{1/L_{\max}, 1/\lambda^{2/3}\})$. Then, for all $t \leq t_0$, the parameters evolve as follows:*

*(1) The dynamics of linear layer $u$ is governed by the following inequalities.*

$$\langle u_t, E_1^w \rangle = \Theta(\eta t), \qquad \forall w,$$
$$\langle u_t, E_\ell^w \rangle = -\Theta(\eta^2 t^2), \qquad \forall \ell \geq 2, \forall w,$$
$$\langle u_t, E_2^w - E_\ell^w \rangle \leq -\Omega(\eta^2 t^2), \quad \forall \ell \geq 3, \forall w.$$

*(2) The dynamics of attention layer $W$ is governed by the following inequalities. For any length $L \leq L_{\max}$, we have*

$$\langle E_1^w - E_\ell^{w'}, W_t E_L^w \rangle \geq \Omega(\eta^2 t^2), \quad \forall \ell \geq 2, \forall w, w'$$
$$\langle E_\ell^{w'} - E_1^{-w}, W_t E_L^w \rangle \geq \Omega(\eta^2 t^2), \quad \forall \ell \geq 2, \forall w, w'$$
$$\langle E_2^{w'} - E_\ell^{w''}, W_t E_L^w \rangle \geq \Omega(\eta^4 t), \quad \forall \ell \geq 3, \forall w, w', w''.$$

Theorem 4.1 characterizes the following featuring dynamics of the linear and attention layers in Phase 1. **(a)** The first equation of part (1) indicates that there is a rapid growth of

the first token score. This is because sequences of length $L = 1$ (which always have positive labels) dominate the early training, and create an initial bias for the first token. **(b)** The first two equations of part (2) indicate that the attention weight of the first token increases in *positive* samples (where last token $E_L^w$ and first token $E_1^w$ share the same value $w$), and is suppressed in *negative* samples. In other words, the attention layer allocates more weights on non-leading tokens (with $\ell \geq 2$) in negative samples. Consequently, the transformer output of the negative samples relies more on the token scores at non-leading positions. In order to minimize the loss, those token scores become increasingly negative, as shown in the second equation of part (1). **(c)** The last equation of part (1) indicates that the token score of the second token decreases faster than other non-leading tokens, as it appears more frequently across samples. This rapid token score decrease drives the attention layer to allocate more attention weight on it over other non-leading tokens to reduce the loss, as shown in the last equation in part (2).

Due to the fact that attention weight is determined by the differences between attention scores, Theorem 4.1 suggests that at the end of Phase 1, the attention layer focuses on the first token (with $\varphi_1^{(n,t_0)} > 1/L$ for length-$L$ samples) in *positive* samples, and on the second token (with $\varphi_2^{(n,t_0)} > 1/L$ for length-$L$ samples) in *negative* samples. Hence, the attention layer maps data samples to satisfy a *separable* property. To be more specific, we first introduce the definition of separable data.

**Definition 4.2** (Separable dataset)**.** A dataset of $d$-dimensional vectors and their labels $\{(v^{(n)}, y_n)\}_{n=1}^N$ are separable if there exists $u \in \mathbb{R}^d$ such that

$$\langle u, y_n v^{(n)} \rangle > 0, \quad \forall 1 \leq n \leq N.$$

Intuitively speaking, a dataset is separable indicates that there exists a hyperplane that can correctly separate the positive and negative samples into two half-spaces.

At the end of Phase 1, the attention layer $W_{t_0}$ maps data samples to $\sum_{\ell=1}^L x_\ell^{(n)} \varphi_\ell^{(n,t_0)}$. It can be shown that if the linear layer has parameter $u = E_1^a + E_1^b - E_2^a - E_2^b$, then, the predicted label of the data sample, which is the sign of $u^\top \sum_{\ell=1}^L x_\ell^{(n)} \varphi_\ell^{(n,t_0)}$, matches with the ground-truth label $y_n$, i.e., $\langle u, y_n \sum_{\ell=1}^L x_\ell^{(n)} \varphi_\ell^{(n,t_0)} \rangle > 0$ for all $n$. Thus, we have the following proposition.

**Proposition 4.3.** *Let $v^{(n)} = \sum_{\ell=1}^L x_\ell^{(n)} \varphi_\ell^{(n,t_0)}$ with label $y_n$. Then, at the end of phase 1, the dataset $\{(v^{(n)}, y_n)\}$ is separable by $u = E_1^a + E_1^b - E_2^a - E_2^b$.*

We note that at the end of Phase 1, the linear layer is trained to be $u_{t_0}$, which may not necessarily separate the attention layer's outputs. In fact, the continual training into Phase 2 will further update the linear layer, so that the attention layer's outputs can be separated by it.

## 4.2. Phase 2: Margin Maximization and Implicit Bias

In this phase, the transformer shifts focus from rapid feature alignment to margin maximization, driven by the implicit bias of gradient descent.

Since the outputs of the attention layer at the end of Phase 1 become separable, we can define the max-margin solution of the corresponding separating hyperplane as follows.

$$u_{EP}^* = \arg \min \|u\|$$
$$\text{s.t. } \left\langle u, y_n \sum_{\ell=1}^{L} x_\ell^{(n)} \varphi_\ell^{(n,t_0)} \right\rangle \geq 1, \quad \forall n \in I.$$

The following theorem characterizes the training dynamics of Phase 2, which shows that the linear layer will converge to the above max-margin solution in direction.

**Theorem 4.4** (Phase 2). *There exists a constant $t_2 = \Omega(1)$ and $T = O(\lambda^{2/3}/(\eta L_{\max}))$, such that for $t_2 \leq t \leq T$, we have $\|u_t\| \geq \Omega(\log t)$, and*

$$\left\langle \frac{u_t}{\|u_t\|}, \frac{u_{EP}^*}{\|u_{EP}^*\|} \right\rangle \geq 1 - \frac{1}{2} \left( \frac{1}{6\|u_{EP}^*\|} - \frac{1}{\|u_t\|} \right)^2.$$

*Moreover, we have $\|W_t - W_{t_0}\| \leq O(1)$.*

Theorem 4.4 characterizes the following dynamics of the linear and attention layers in Phase 2. **(a)** The norm $\|u_t\|$ of the linear layer continues to grow logarithmically to increase the classification margin. **(b)** The updates of the attention layer $W_t$ is negligible due to the scaling factor $\lambda$, as indicated by $\|W_t - W_{t_0}\| \leq O(1)$. Hence, attention patterns at the end of Phase 1 persist, i.e., in both positive and negative samples, the attention scores of the first and second tokens still dominate, respectively. **(c)** The linear layer $u$ enters a regime governed by implicit bias, which converges to the max-margin solution for separating the attention layer's outputs. This dynamics is also observed in an empirical work (Merrill et al., 2021).

**Theorem 4.5** (Convergence of loss). *For $t \leq T$, we have $\mathcal{L}_t = O\left( \frac{L_{\max}\|u_{EP}^*\|^2}{\eta\sqrt{t}} \right)$.*

Theorem 4.5 indicates that the loss converges to $O\left( \frac{\eta^{1/2}}{\lambda^{1/3}} \right)$. Therefore, as long as the scaling factor $\lambda = \Omega(\frac{\eta^{2/3}}{\epsilon^3})$, the loss can achieve arbitrarily small value $\epsilon$. The full proof in this section can be found in Appendix B.

In summary, the trained transformer utilizes its attention to decide if two tokens are equal and the linear layer increases the classification margin and enable fast loss decay.

## 5. Parity Check Problem

The parity check problem is generally considered to be more difficult than the even pairs problem. For instance, it has

been shown in Pérez et al. (2021) that it is impossible to recognize parity by applying transformer once. However, it has recently been shown in Kim & Suzuki (2024b) that chain-of-thought (CoT) can serve as an advanced approach to solving such a task.

In this section, we provide two new approaches to solving the parity check problem, by integrating CoT with the solution for the even pairs problem studied in Section 4. The first approach solves parity check by applying the one-layer transformer well-trained for even pairs via a truncated CoT-type *inference*, which does not require any additional training. The second approach trains a one-layer transformer with CoT under teacher forcing, where GD provably renders a transformer that solves the parity check problem.

### 5.1. Approach 1: Inference via Truncated CoT

Inspired by the 2-state machine, we show that by simply taking *inference* via truncated CoT, the one-layer transformer well-trained for the even pairs problem can solve parity check efficiently without additional training.

To formalize this, we first outline the method to solve parity check through the lens of a 2-state finite automaton. Recall that the parity check problem is to determine whether the number of bs in a binary string $X \in \{a, b\}^L$ is even or odd. Given an $X = w_1 w_2 \cdots w_L$, the automaton initializes its state $s_1$ to $w_1$ and updates $s_t$ for $t \geq 2$ sequentially as follows. At each step $t \geq 2$, the state transits to $s_{t+1} = a$ if $s_t = w_{t+1}$, or $s_{t+1} = b$ if $s_t \neq w_{t+1}$. For example, for $X = abb$, the state transitions are $s_1 = a \rightarrow s_2 = b$ (as $s_1 \neq w_2 = b$) $\rightarrow s_3 = a$ (as $s_2 = w_3 = b$). The character a of the last state indicates that the sequence takes even parity (we equate 1 with token a and $-1$ with token b). It is worth noting that the core step of parity check involves comparing two characters each time, which is also performed in even pairs except that the characters to be checked are fixed to be the first and last tokens.

Inspired by this observation, we propose the following inference method via truncated CoT to solve parity check.

**Inference via truncated CoT.** Recall that the even pairs problem is equivalent to labeling whether the first and the last tokens are the same. Hence, the transformer trained for even pairs can provide correct labels for such a task during the inference. We thus propose truncated CoT that leverages the label predicted by the trained transformer for even pairs iteratively to obtain the answer for parity check. Such an *inference* process runs as follows, with the pseudocode provided in Algorithm 1. Given a binary sequence $X = w_1 \cdots w_L$, at each iteration $t \in \{1, \ldots, L-1\}$ of CoT, it performs the following steps: (1) check whether the first and the last tokens of $X$ are equal by applying the one-layer transformer trained for even pairs; (2) append the predicted

**Algorithm 1** Truncated CoT

---

**Input:** Binary sequence $X = w_1 \cdots w_L$
**for** $t \in \{1, \ldots, L-1\}$ **do**
    Predict $y_t = T_{\theta_T}(X)$
    Let $w_{L+t} = $ a if $y_t = 1$ and $w_{L+t} = $ b if $y_t = -1$.
    Update $X = w_{t+1} \cdots w_{L+t}$.
**end for**
**Output** $y_{L-1}$.

---

label $y = $ a or $y = $ b to the end of the sequence $X$ (we equate 1 with token a and $-1$ with token b); (3) remove the first token in $X$ to maintain the sequence length. Then after $L-1$ iterations, the final prediction provides the parity of the original input sequence.

### 5.2. Approach 2: Training with CoT under Teacher Forcing

In this section, we train a one-layer transformer with the full version of Chain-of-Thought (CoT) to solve the parity check problem. Unlike the truncated CoT described in Section 5.1, we keep the original sequence and only append the predicted label to the end of the sequence at each iteration.

To explain how we train a one-layer transformer to execute CoT learning of parity, we first describe how the parity of a sequence can be obtained via CoT step by step.

For a given sequence $X = w_1 \cdots w_{L_0}$ with length $L_0$, we generate CoT inputs $X^1, \ldots, X^{L_0-1}$ and their corresponding labels to learn the parity as follows. First, let $X^1 = X$. For each $t \in \{1, \ldots, L_0 - 1\}$, take $X^t$ and compare its tokens $w_{L_0+t-1}$ and $w_t$. If they are the same, let $w_{L_0+t} = $ a; otherwise, let $w_{L_0+t} = $ b. Then set $w_{L_0+t}$ to be the label of $X^t$, and append $w_{L_0+t}$ to the end of $X^t$ to obtain $X^{t+1} = w_1 \cdots w_{L_0+t}$ for the next step of CoT. Finally, label $w_{2L_0-1}$ is the parity of $X$ (See Figure 1).

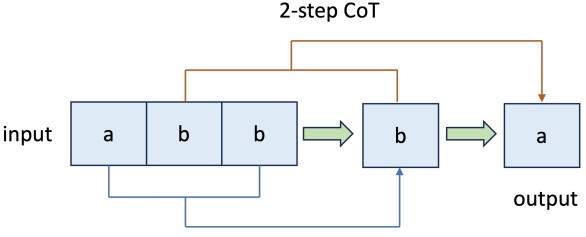

Figure 1. An illustration of CoT on input abb, where $L_0 = 3$.

**Training design.** To train a transformer to learn each step $1 \le t \le L_0 - 1$ of CoT, our dataset should be labeled to compare the token $w_{L_0+t-1}$ (last token of the input sequence of the current CoT step) and $w_t$, and such a label is used to train (supervise) the transformer to conduct the $t$-th step of CoT correctly. Thus, to train the $t$-th step of CoT, we

use the set $I_L$ of length-$L$ sequences (where $L = L_0+t-1$), and for each sequence $X^{(n)} \in I_L$, set its label $y_n^p$ to be 1 if the last token matches the token at position $L - L_0 + 1 = t$, and $-1$ otherwise. The superscript '$p$' in $y_n^p$ indicates that the label is constructed for parity check. Thus, the training of CoT will use data sequences with length $L$ where $L_0 \le L \le 2L_0 - 1$ and the total loss is given by

$$\mathcal{L}_{CoT}(u, W)$$
$$= \sum_{L=L_0}^{2L_0-1} \frac{1}{|I_L|} \sum_{n \in I_L} \log\left(1 + \exp\left(-y_n^p T_\theta(X^{(n)})\right)\right),$$

where we adopt the one-layer transformer described in Section 3 with the dimension $d \ge 2L_0 - 2$.

Furthermore, we observe in our experiments that if we directly train transformers over the above CoT loss $\mathcal{L}_{CoT}(u, W)$, the gradient vanishes. Interestingly, initializing transformer by that trained for even pairs helps to avoid such a case. Motivated by such an observation, we introduce the even pairs loss to regularize the training process. To this end, we include data sequences with length $L < L_0$ for the even pairs loss, and label those sequences by their even pairs labels. Namely, for any sequence $X^{(n)}$, where $n \in I_L$ with $L \le L_0$, set the label $y_n^e$ to be 1 if the last token matches the first token, and $-1$ otherwise. The superscript '$e$' in $y_n^e$ indicates that the label is constructed for even pairs. These data sequences provide a regularization loss given by

$$\mathcal{L}_{Reg}(u, W)$$
$$= \sum_{L=1}^{L_0-1} \frac{1}{|I_L|} \sum_{n \in I_L} \log\left(1 + \exp\left(-y_n^e T_\theta(X^{(n)})\right)\right).$$

The role of the regularization loss is to initially guide the linear layer to rapidly increase along the direction of solving even pairs problems, which will also initialize the parameters to learn parity check in a stable way. We note that such regularization is also equivalent to data mixing technique.

Hence, the total training loss for parity check is given by

$$\mathcal{L}_{Parity} = \mathcal{L}_{CoT}(u, W) + \mathcal{L}_{Reg}(u, W). \tag{1}$$

We minimize the above loss function by gradient descent as described in Section 3 for training the parity check problem. In the following, we choose $I_L = \{$a,b$\}^L$ for simplicity.

### 5.3. Training Dynamics of Approach 2

The training process of CoT under teacher forcing can also be divided into two training phases. Below, we present the theoretical characterization of those two phases.

**Theorem 5.1** (Phase 1). *Let $-w$ denote the flip of token $w \in \{a, b\}$. Choose $\lambda = \Omega(L_{\max}^2)$ and $t_0 = O(1/(\eta L_{\max}))$. Then, for all $t \le t_0$, the parameters evolve as follows:*

*(1) The dynamics of linear layer $u$ is governed by the fol-*

*lowing inequalities.*

$$\langle u_t, E_1^w \rangle = \Theta(\eta t), \quad \forall w,$$
$$\langle u_t, E_\ell^w \rangle = -\Theta(\eta^2 t^2), \quad \forall \ell \geq 2, \forall w,$$
$$\langle u_t, E_2^w - E_\ell^w \rangle \leq -\Omega(\eta^2 t^2), \quad \forall \ell \geq 3, \forall w.$$

*(2) The dynamics of the attention layer $W$ is governed by the following inequalities. For any length $L < L_0$, we have*

$$\langle E_1^w - E_\ell^{w'}, W_t E_L^w \rangle \geq \Omega(\eta^2 t^2), \quad \forall \ell \geq 2, \forall w, w',$$
$$\langle E_\ell^{w'} - E_1^{-w}, W_t E_L^w \rangle \geq \Omega(\eta^2 t^2), \quad \forall \ell \geq 2, \forall w, w',$$
$$\langle E_2^{w'} - E_\ell^{w''}, W_t E_L^w \rangle \geq \Omega(\eta^4 t), \quad \forall \ell \geq 3, \forall w, w', w''.$$

*For length $L \geq L_0$, let $\ell_0 = L - L_0 + 1$, and we have*

$$\langle E_{\ell_0}^{-w} - E_\ell^{w'}, W_t E_L^w \rangle \geq \Omega(\eta^2 t^2), \quad \forall \ell \neq \ell_0, \forall w, w',$$
$$\langle E_\ell^{w'} - E_{\ell_0}^{-w}, W_t E_L^w \rangle \geq \Omega(\eta^2 t^2), \quad \forall \ell \neq \ell_0, \forall w, w',$$
$$\langle E_1^{w'} - E_\ell^{w''}, W_t E_L^w \rangle \geq \Omega(\eta^4 t), \quad \forall \ell \neq 1, \ell_0, \forall w, w', w''.$$

Since the loss function in Equation (1) is regularized by the loss of even pairs, the linear layer $u$ exhibits the same dynamics as the even pairs problem in Theorem 4.1. The key difference between CoT training in Theorem 5.1 and even pairs training in Theorem 4.1 lies in attention dynamics on sequences with length $L \geq L_0$, which are labeled for CoT training of parity check. In particular, the last three inequalities in Theorem 5.1 suggests that it is the token at position $L - L_0 + 1$ that differs most from other tokens.

At the end of Phase 1, the outputs of the attention layer are also separable. Namely, there exists a linear classifier that provides correct labels for all CoT steps, i.e., labels all training sequences with length $L_0 \leq L \leq 2L_0 - 1$ correctly. As a by-product, such a linear classifier also provides correct even pairs labels for the sequences with $L < L_0$. For those separable data sequences, we define thee max-margin solution for the separating hyperplane as

$$u_{CoT}^* = \arg\min \|u\|,$$
$$\text{s.t. } \left\langle u, y_n \sum_{\ell=1}^{L(X^{(n)})} x_\ell^{(n)} \varphi_\ell^{(n,t_0)} \right\rangle \geq 1, \quad \forall n \in I,$$

where $L(X^{(n)})$ denotes the length of $X^{(n)}$. Note that $u_{CoT}^*$ slightly abuses notation as the dataset also includes sequences with lengths $L < L_0$ for even pairs.

The following theorem shows that the training enters Phase 2 if we continue to update the parameters by gradient descent, during which the attention layer has negligible change, but the linear layer converges to the max-margin solution $u_{CoT}^*$.

**Theorem 5.2** (Phase 2). *There exists a constant $t_2 = \Omega(1)$ and $T = O(\lambda^{2/3}/(\eta L_{\max}))$, such that for $t_2 \leq t \leq T$, we*

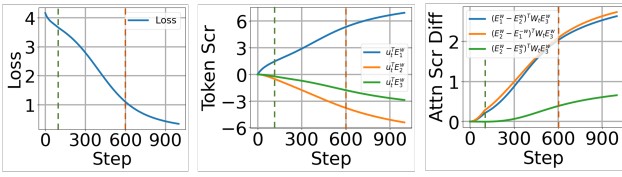

*Figure 2.* Results of one-layer transformer on even pairs. From the left to the right: (1) Loss decay over training. (2) Token scores at first three positions. (3) Attention scores in length-3 sequences.

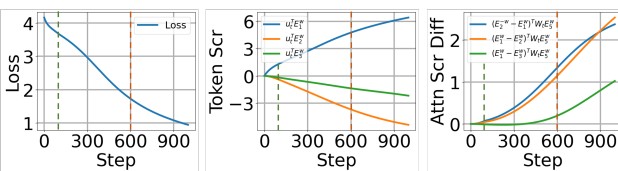

*Figure 3.* Results of one-layer transformer on parity check. From the left to the right: (1) Loss decay over training. (2) Token scores at first three positions. (3) Attention scores in length-5 sequences.

*have $\|u_t\| \geq \Omega(\log t)$, and*

$$\left\langle \frac{u_t}{\|u_t\|}, \frac{u_{CoT}^*}{\|u_{CoT}^*\|} \right\rangle \geq 1 - \frac{1}{2} \left( \frac{1}{6\|u_{CoT}^*\|} - \frac{1}{\|u_t\|} \right)^2.$$

*Moreover, we have $\|W_t - W_{t_0}\| \leq O(1)$.*

The above theorem also implies that the total loss converges sublinearly, which further implies that both the CoT and regularization losses enjoy the same decay rate.

**Theorem 5.3.** *For $t \leq T$, we have $\mathcal{L}_{Parity,t} = O\left( \frac{L_{\max}\|u_{CoT}^*\|^2}{\eta\sqrt{t}} \right)$.*

# 6. Experiments

In this section, we provide experiments on synthetic datasets to verify our theoretical findings. Specifically, we choose $L_{\max} = 6$, $L_0 = 4$. Then, we train $u$ and $W$ by gradient descent with step size $\eta = 0.1$. We choose $t_0 = 100$, and $\lambda = 2$. As observed in Figures 2 and 3, the first plot in each figure shows the rapid decay of the loss to the global minimum. The second plot shows the dynamics of token scores at the first three positions, where the first token score grows (blue curve) and the second and third token scores decrease (green and orange curves) during training. The third plot illustrates dynamics of the attention weights, where the first token receives more attention in positive samples (blue curve) and less attention in negative samples (orange curve). These plots validate our theoretical findings on dynamics of token scores and attention scores characterized in Theorem 4.1 and Theorem 5.1. Furthermore, the vertical blue dashed line in these plots indicates the end of first phase. The vertical orange dashed line in these plots

indicates $t_2 \approx 600$ in Theorem 4.4 and Theorem 5.2, where $\|u_t\|$ starts to grow logarithmically (second plot of both figures).

**Additional Experiments.** In Appendix D, we conduct more experiments on different configurations of scaling parameter $\lambda$ and the two-phase learning dynamics.

All experiments are conducted on a PC equipped with an i5-12400F processor and 16GB of memory.

## 7. Conclusion

In this work, we provide a theoretical characterizing of two training phases to uncover how a one-layer transformer can be trained to solve two regular language recognition problems: even pairs and parity check. In order to characterize the joint training of attention and linear layers, we employ higher-order Taylor expansions to precisely analyze the coupling between gradients of two layers and its impact on parameter updates. Our results not only offer deeper insights into the training behavior of transformers but also highlight the critical role of CoT in solving parity problems. Experimental validation further supports our theoretical findings, confirming key aspects of parameter evolution and convergence. The analysis tools developed in this work can be useful for future understanding the implicit biases and training dynamics of transformers in structured learning tasks.

## Acknowledgements

The work of Y. Liang was supported in part by the U.S. National Science Foundation under the grants ECCS-2113860 and DMS-2134145. The work of R. Huang and J. Yang was supported in part by the U.S. National Science Foundation under the grants CNS-1956276 and CNS-2114542.

## Impact Statement

This paper presents work whose goal is to advance the field of Machine Learning. There are many potential societal consequences of our work, none which we feel must be specifically highlighted here.

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

# A. Auxiliary Lemmas and Equations

**Lemma A.1** (Gao & Pavel (2017)). *The softmax function with scaling factor $\lambda$ is $\frac{1}{\lambda}$-Lipschitz continuous. Mathematically, we have*

$$\|\phi(x/\lambda) - \phi(y/\lambda)\| \leq \frac{1}{\lambda}\|x - y\|.$$

By Lemma A.1 and $\varphi_\ell^{(n,t)} = \frac{\exp\left(\langle x_\ell^{(n)}, W_t x_L^{(n)}\rangle/\lambda\right)}{\sum_{\ell'} \exp\left(\langle x_{\ell'}^{(n)}, W x_L^{(n)}\rangle/\lambda\right)}$, we have the following inequalities, which will be itensively used in the subsequence proofs.

$$|\varphi_\ell^{(n,t)} - \varphi_\ell^{(n,t')}| \leq \sqrt{\sum_{\ell=1}^{L} |\varphi_\ell^{(n,t)} - \varphi_\ell^{(n,t')}|^2} \leq \frac{\|W_t - W_{t'}\|}{\lambda}$$

Let $J(x) = \log(1 + e^{-x})$ be the logistic loss. We also frequently use the following Taylor expansion about $-J'(x)$.

$$|J(x)' - J(x')'| \leq |x - x'|, \quad \left|\frac{1}{1 + \exp(x)} - \left(\frac{1}{2} - \frac{1}{4}x\right)\right| \leq x^3. \tag{2}$$

## A.1. Gradients Calculation

Recall that the loss at time step $t$ is

$$\mathcal{L}_t = \sum_{L=1}^{L_{\max}} \frac{1}{2^L} \sum_{n \in I_L} J\left(y_n u_t^\top \sum_\ell x_\ell^{(n)} \varphi_\ell^{(n,t)}\right).$$

For simplicity, we use the following notation. For each sample $n$ and time step $t$, we define

$$J'_{(n,t)} = -\frac{1}{1 + \exp(y_n \mathtt{T}_t(X^{(n)}))},$$

where $\mathtt{T}_t(X^{(n)}) = u_t^\top \sum_\ell x_\ell^{(n)} \varphi_\ell^{(n,t)}$. Therefore, the gradients can be written as follows. The gradients at time $t$ are

$$\begin{cases} \nabla_u \mathcal{L}_t = \sum_{L=1}^{L_{\max}} \frac{1}{2^L} \sum_{n \in I_L} J'_{(n,t)} y_n \sum_\ell x_\ell^{(n)} \varphi_\ell^{(n,t)} \\ \nabla_W \mathcal{L}_t = \frac{1}{\lambda} \sum_{L=2}^{L_{\max}} \frac{1}{2^L} \sum_{n \in I_L} J'_{(n,t)} y_n \sum_\ell u_t^\top x_\ell^{(n)} \varphi_\ell^{(n,t)} \left(x_\ell^{(n)} - \sum_{\ell'} \varphi_{\ell'}^{(n,t)} x_{\ell'}^{(n)}\right) (x_L^{(n)})^\top \end{cases}$$

Note that $\|\nabla_u \mathcal{L}_t\| \leq L_{\max}$ and $\|\nabla_W \mathcal{L}_t\| \leq \|u_t\| L_{\max}/\lambda$. We will also frequently use the following projection of gradients on each token embeddings.

$$\begin{cases} \langle \nabla_u \mathcal{L}_t, E_\ell^w \rangle = \sum_{L \geq \ell} \frac{1}{2^L} \sum_{n \in I_L, \mathtt{C}(x_\ell^{(n)}) = w} J'_{(n,t)} y_n \varphi_\ell^{(n,t)}, \\ \langle E_1^w, (\nabla_W \mathcal{L}_t) E_L^w \rangle = \frac{1}{2^L} \sum_{n \in I_L, \mathtt{C}(x_1^{(n)}) = \mathtt{C}(x_L^{(n)}) = w} J'_{(n,t)} \varphi_1^{(n,t)} \left(u_t^\top x_1^{(n)} - \mathtt{T}(X^{(n)})\right), \\ \langle E_1^{-w}, (\nabla_W \mathcal{L}_t) E_L^w \rangle = -\frac{1}{2^L} \sum_{n \in I_L, \mathtt{C}(x_1^{(n)}) \neq \mathtt{C}(x_L^{(n)}) = w} J'_{(n,t)} \varphi_1^{(n,t)} \left(u_t^\top x_1^{(n)} - \mathtt{T}(X^{(n)})\right), \\ \langle E_\ell^w, (\nabla_W \mathcal{L}_t) E_L^w \rangle = \frac{1}{2^L} \sum_{n \in I_L, \mathtt{C}(x_\ell^{(n)}) = \mathtt{C}(x_L^{(n)}) = w} J'_{(n,t)} y_n \varphi_\ell^{(n,t)} \left(u_t^\top x_\ell^{(n)} - \mathtt{T}(X^{(n)})\right), \end{cases}$$

where $\mathtt{C}(x_\ell^{(n)})$ is the character of token embedding $x_\ell^{(n)}$. For example, $\mathtt{C}(E_\ell^w) = w$.

## B. Proofs of Even Pairs Problem

In this section, we provide the full proof of training dynamics of transformers on even pairs problem. Before proceeding to analyzing phase 1 and phase 2, we first show that the token scores only depend on the position, which helps us to reduce the complexity of the subsequent proof.

**Lemma B.1.** *During the entire training process, for any $w \in \{a, b\}$, we have*

$$\langle u_t, E_\ell^w \rangle = \langle u_t, E_\ell^{-w} \rangle, \quad \forall \ell \geq 1.$$

*For attention scores, we have the following equalities.*

$$\begin{cases} \langle E_1^w, W_t E_L^w \rangle = \langle E_1^{-w}, W_t E_L^{-w} \rangle. \\ \langle E_\ell^w, W_t E_L^w \rangle = \langle E_\ell^{-w}, W_t E_L^w \rangle, \quad \forall \ell \geq 2. \end{cases}$$

*Proof.* Note that the results are valid for $t = 0$. Assume the results hold at time $t$, we aim to prove the results hold for $t + 1$. It suffices to prove the following equalities.

$$\langle \nabla_u \mathcal{L}_t, E_\ell^w \rangle = \langle \nabla_u \mathcal{L}_t, E_\ell^{-w} \rangle, \quad \forall \ell \geq 1, \tag{3}$$

$$\langle E_\ell^w, (\nabla_W \mathcal{L}_t) E_L^w \rangle = \langle E_\ell^{-w}, (\nabla_W \mathcal{L}_t) E_L^w \rangle, \quad \forall \ell \geq 2, \tag{4}$$

$$\langle E_1^w, (\nabla_W \mathcal{L}_t) E_L^w \rangle = \langle E_1^{-w}, (\nabla_W \mathcal{L}_t) E_L^{-w} \rangle. \tag{5}$$

**We first show that Equation (3) is true.**

For $\ell \geq 2$, we have

$$\langle \nabla_u \mathcal{L}_t, E_\ell^w \rangle = \sum_{L \geq \ell} \sum_{n \in I_L, \mathtt{C}(x_\ell^{(n)}) = w} J'_{(n,t)} y_n \varphi_\ell^{(n,t)}$$

$$= \sum_{L \geq \ell} \sum_{n \in I_L, \mathtt{C}(x_\ell^{(n)}) = w} \frac{-1}{1 + \exp(y_n \mathtt{T}_t(X_L^{(n)}))} y_n \varphi_\ell^{(n,t)}.$$

For any $n \in I_L$ satisfying $\mathtt{C}(x_\ell^{(n)}) = w$, let $n'$ be the sample that only replace $w$ with $-w$ at the $\ell$-th position. Then, due to the induction hypothesis, we have $\varphi_\ell^{(n,t)} = \varphi_\ell^{(n',t)}$, and $\mathtt{T}_t(X_L^{(n)}) = \mathtt{T}_t(X_L^{(n')})$. Since $\ell \geq 2$, changing one token at the position $\ell$ does not change the label, we have $y_n = y_{n'}$. Therefore, we have

$$\langle \nabla_u \mathcal{L}_t, E_\ell^w \rangle = \langle \nabla_u \mathcal{L}_t, E_\ell^{-w} \rangle, \quad \forall \ell \geq 2.$$

For $\ell = 1$, we have,

$$\langle \nabla_u \mathcal{L}_t, E_1^w \rangle = \sum_{L \geq 1} \sum_{n \in I_L, \mathtt{C}(x_1^{(n)}) = w} J'_{(n,t)} y_n \varphi_1^{(n,t)}$$

$$= \sum_{L \geq 1} \sum_{n \in I_L, \mathtt{C}(x_1^{(n)}) = w} \frac{-1}{1 + \exp(y_n \mathtt{T}_t(X_L^{(n)}))} y_n \varphi_1^{(n,t)}$$

Now, for any $n \in I_L$ satisfying $\mathtt{C}(x_1^{(n)}) = w$, let $n'$ be the sample that flips the first and the last token at the same time. Then, due to the induction hypothesis, we have $\varphi_\ell^{(n,t)} = \varphi_\ell^{(n',t)}$, $y_n = y_{n'}$ and $\mathtt{T}_t(X_L^{(n)}) = \mathtt{T}_t(X_L^{n'})$. Therefore, we have

$$\langle \nabla_u \mathcal{L}_t, E_1^w \rangle = \langle \nabla_u \mathcal{L}_t, E_1^{-w} \rangle.$$

**We conclude that Equation (3) is true.**

**Then, we show that Equation (4) is true.** For $\ell \geq 2$, we have

$$\langle E_\ell^w, (\nabla_W \mathcal{L}_t) E_L^w \rangle = \frac{1}{2^L} \sum_{n \in I_L, \mathtt{C}(x_\ell^{(n)}) = \mathtt{C}(x_L^{(n)}) = w} J'_{(n,t)} y_n \varphi_\ell^{(n,t)} \left( u_t^\top x_\ell^{(n)} - \mathtt{T}_t(X_L^{(n)}) \right)$$

$$= \frac{1}{2^L} \sum_{n \in I_L, \mathtt{C}(x_\ell^{(n)}) = \mathtt{C}(x_L^{(n)}) = w} \frac{-1}{1 + \exp\left(y_n \mathtt{T}_t(X^{(n)})\right)} y_n \varphi_\ell^{(n,t)} \left( u_t^\top x_\ell^{(n)} - \mathtt{T}_t(X_L^{(n)}) \right)$$

For any $n \in I_L$ satistyfing $\mathtt{C}(x_\ell^{(n)}) = \mathtt{C}(x_L^{(n)}) = w$, let $n'$ be the sample that only replace $w$ with $-w$ at the $\ell$-th position. Then, due to the induction hypothesis, we have $\varphi_\ell^{(n,t)} = \varphi_\ell^{(n',t)}$, and $\mathtt{T}_t(X_L^{(n)}) = \mathtt{T}_t(X_L^{(n')})$. Since $\ell \geq 2$, changing one token at the position $\ell$ does not change the label, we have $y_n = y_{n'}$. Therefore, we have

$$\langle E_\ell^w, (\nabla_W \mathcal{L}_t) E_L^w \rangle = \langle E_\ell^{-w}, (\nabla_W \mathcal{L}_t) E_L^w \rangle, \quad \forall \ell \geq 2.$$

**We conclude that Equation (4) is true.**

**Finally, we show that Equation (5) is true.**

Note that

$$\langle E_1^w, (\nabla_W \mathcal{L}_t) E_L^w \rangle = \frac{1}{2^L} \sum_{n \in I_L, \mathtt{C}(x_1^{(n)}) = \mathtt{C}(x_L^{(n)}) = w} J'_{(n,t)} \varphi_1^{(n,t)} \left( u_t^\top x_1^{(n)} - \mathtt{T}_t(X_L^{(n)}) \right)$$

$$= \frac{1}{2^L} \sum_{n \in I_L, \mathtt{C}(x_1^{(n)}) = \mathtt{C}(x_L^{(n)}) = w} \frac{-1}{1 + \exp\left(y_n \mathtt{T}_t(X_L^{(n)})\right)} \varphi_1^{(n,t)} \left( u_t^\top x_1^{(n)} - \mathtt{T}_t(X_L^{(n)}) \right)$$

Now, for any $n \in I_L$ satisfying $\mathtt{C}(x_1^{(n)}) = \mathtt{C}(x_L^{(n)}) = w$, let $n'$ be the sample that flips the first and the last token at the same time, i.e., $\mathtt{C}(x_1^{(n')}) = \mathtt{C}(x_L^{(n')}) = -w$. Then, due to the induction hypothesis, we have $\varphi_\ell^{(n,t)} = \varphi_\ell^{(n',t)}$, $y_n = y_{n'}$ and $\mathtt{T}_t(X_L^{(n)}) = \mathtt{T}_t(X_L^{(n')})$. Therefore, we have

$$\langle \nabla_u \mathcal{L}_t, E_1^w \rangle = \langle \nabla_u \mathcal{L}_t, E_1^{-w} \rangle.$$

**We conclude that Equation (5) is true.**

Therefore, the proof is complete by induction. $\qquad \square$

Due to above lemma, for each length $L$, we only need to analyze two types of sequence, i.e., the one with positive label and the one with negative label. We use $X_L^{(+)} = [E_1^w, E_2^w, \ldots, E_L^w s]$ to represent the sequence with positive labels, and $X_L^{(-)} = [E_1^{-w}, E_2^w, \ldots, E_L^2]$ to represent the sequence with negative labels.

### B.1. Phase 1

In this section, we characterize the training dynamics in phase 1. In general, we prove the results by induction. First, we characterize the initialization dynamics.

**Lemma B.2** (Initialization). *At the beginning ($t = 2, 3$), for the linear layer, we have*

$$\langle u_2, E_1^w \rangle = \Theta(\eta t), \quad \forall w,$$
$$\langle u_2, E_\ell^w \rangle = -\Theta(\eta^2 t^2), \quad \forall \ell \geq 2, \forall w,$$
$$\langle u_2, E_2^w - E_\ell^w \rangle \leq -\Omega(\eta^2 t^2), \quad \forall \ell \geq 3, \forall w.$$

*For the attention layer, we have*

$$\langle E_1^w - E_2^w, W_3 E_L^w \rangle \geq \Omega\left(\frac{\eta^2}{L}\right), \quad \forall w,$$

$$\left\langle E_1^{-w} - E_2^w, W_3 E_L^w \right\rangle \leq -\Omega\left(\frac{\eta^2}{L}\right), \quad \forall w,$$

$$\left\langle E_2^w - E_\ell^w, W_3 E_L^w \right\rangle \geq \Omega\left(\frac{\eta^2}{L}\right), \quad \forall \ell \geq 3, \forall w.$$

*Proof.* Since $J_{(n,0)} = \frac{1}{2}$, and $\varphi_\ell^{(n,0)} = \frac{1}{L}$ for $n \in I_L$ (the length is $L$), we have

$$\langle u_1, E_1^w \rangle = \eta \langle -\nabla_u \mathcal{L}_0, E_1^w \rangle$$

$$= \eta \sum_{L=1}^{L_{\max}} \frac{1}{2^L} \sum_{n \in I_L, \mathtt{C}(x_\ell^{(n)})=w} (-J'_{(n,0)}) y_n \varphi_1^{(n,0)}$$

$$= \frac{\eta}{4},$$

where the last equality is due to the cancellation between positive and negative samples whose length is greater than 1. Due to the same reason, for any $\ell \geq 2$, we have

$$\langle u_1, E_\ell^w \rangle = \eta \langle -\nabla_u \mathcal{L}_0, E_\ell^w \rangle$$

$$= \eta \sum_{L=2}^{L_{\max}} \frac{1}{2^L} \sum_{n \in I_L, \mathtt{C}(x_\ell^{(n)})=w} (-J'_{(n,0)}) y_n \varphi_1^{(n,0)}$$

$$= 0$$

Regarding the attention, for any token $E_\ell^{w'}$, we have

$$\langle E_\ell^{w'}, W_1 E_L^w \rangle = \eta \lambda \langle E_\ell^2, (-\nabla_W \mathcal{L}_1) E_L^w \rangle$$

$$= \frac{1}{2^L} \sum_{n \in I_L, \mathtt{C}(x_\ell^{(n)})w', \mathtt{C}(x_L^{(n)})=w} J'_{(n,0)} y_n \varphi_\ell^{(n,0)} \left( u_0^\top x_\ell^{(n)} - \mathtt{T}_0(X^{(n)}) \right)$$

$$= 0,$$

where the last inequality is due to the fact that $u_0 = 0$.

In summary, at time step 1, only the token score at the first position increases, and all other token scores remain 0, and the attention scores are all 0, resulting $\varphi_\ell^{(n,1)} = \frac{1}{L}$ for $n \in I_L$. Note that, we also have

$$-J'_{(n,1)} = \frac{1}{1+\exp(\frac{\eta}{4L})}, n \in I_L^+; \quad -J'_{(n,1)} = \frac{1}{1+\exp(-\frac{\eta}{4L})}, n \in I_L^-.$$

**Next, we characterize the token scores and attention scores at time step 2.**

$$\langle u_2, E_1^w \rangle = \langle u_1, E_1^w \rangle + \eta \langle -\nabla_u \mathcal{L}_1, E_1^w \rangle$$

$$= \frac{\eta}{4} + \eta \sum_{L=2}^{L_{\max}} \frac{1}{2^L} \sum_{n \in I_L, \mathtt{C}(x_\ell^{(n)})=w} (-J'_{(n,0)}) y_n \varphi_1^{(n,0)}$$

$$= \frac{\eta}{4} + \frac{\eta}{2} \frac{1}{1+\exp(\eta/4)} + \sum_{L=2}^{L_{\max}} \frac{\eta}{2} \left( \frac{1}{1+\exp(\frac{\eta}{4L})} - \frac{1}{1+\exp(-\frac{\eta}{4L})} \right) \frac{1}{L}$$

Thus,

$$\left| \langle u_2, E_1^w \rangle - \frac{2\eta}{4} \right| \leq \frac{\eta}{2} \left| \frac{1}{1+\exp(\eta/4)} - \frac{1}{2} \right| + \sum_{L=2}^{L_{\max}} \frac{\eta}{2L} \cdot \frac{\eta}{2L}$$

$$\leq \frac{3\eta^2}{8},$$

where the last inequality is due to the Lipchitz continuity of $f(x) = 1/(1+e^x)$ (with Lipchitz constant 1) and $\sum_{L=2}^{\infty} 1/L^2 \leq 1$.

Similarly, for $\ell \geq 2$, we have

$$\langle u_2, E_\ell^w \rangle = \eta \langle -\nabla_u \mathcal{L}_1, E_\ell^w \rangle$$

$$= \eta \sum_{L=\ell}^{L_{\max}} \frac{1}{2^L} \sum_{n \in I_L, \mathtt{C}(x_\ell^{(n)})=w} (-J_{(n,1)}') y_n \varphi_\ell^{(n,0)}$$

$$= \sum_{L=\ell}^{L_{\max}} \frac{\eta}{2} \left( \frac{1}{1 + \exp(\frac{\eta}{4L})} - \frac{1}{1 + \exp(-\frac{\eta}{4L})} \right) \frac{1}{L}$$

$$\leq - \sum_{L=\ell}^{L_{\max}} \frac{\eta^2}{4L^2}$$

$$\leq - \frac{\eta^2}{4\ell^2}.$$

In addition, regarding the difference of token scores, we have

$$\langle u_2, E_2^w - E_\ell^2 \rangle = \sum_{L=2}^{L_{\max}} \frac{\eta}{2} \left( \frac{1}{1 + \exp(\frac{\eta}{4L})} - \frac{1}{1 + \exp(-\frac{\eta}{4L})} \right) \frac{1}{L} - \sum_{L=\ell}^{L_{\max}} \frac{\eta}{2} \left( \frac{1}{1 + \exp(\frac{\eta}{4L})} - \frac{1}{1 + \exp(-\frac{\eta}{4L})} \right) \frac{1}{L}$$

$$= \sum_{L=2}^{\ell-1} \frac{\eta}{2} \left( \frac{1}{1 + \exp(\frac{\eta}{4L})} - \frac{1}{1 + \exp(-\frac{\eta}{4L})} \right) \frac{1}{L}$$

$$\leq - \frac{\eta^2}{16},$$

where $\ell \geq 3$.

Therefore, we already prove that

$$\langle u_2, E_1^w \rangle = \Theta(\eta t), \quad \forall w,$$
$$\langle u_2, E_\ell^w \rangle = -\Theta(\eta^2 t^2), \quad \forall \ell \geq 2, \forall w,$$
$$\langle u_2, E_2^w - E_\ell^w \rangle \leq -\Omega(\eta^2 t^2), \quad \forall \ell \geq 3, \forall w.$$

**Next, we analyze the attention score.**

$$\lambda \langle E_1^w - E_2^w, (-\nabla_W \mathcal{L}_1) E_L^w \rangle = \frac{1}{2^L} \sum_{n \in I_L^+} (-J_{(n,1)}') \varphi_1^{(n,1)} \left( \langle u_1, E_1^w \rangle - \frac{\eta}{4L} \right)$$

$$- \frac{1}{2^L} \sum_{n \in I_L, \mathtt{C}(x_2^{(n)})=w} (-J_{(n,1)}') y_n \varphi_2^{(n,1)} \left( \langle u_t, E_2^w \rangle - \frac{\eta}{4L} \right)$$

$$= \frac{1}{8} \cdot \frac{1}{1 + \exp(\frac{\eta}{4L})} \cdot \frac{1}{L} \left[ 2\langle u_1, E_1^w \rangle - \langle u_1, E_2^w \rangle \right]$$

$$+ \frac{1}{8} \frac{1}{1 + \exp(-\frac{\eta}{4L})} \cdot \frac{1}{L} \left( \langle u_1, E_2^w \rangle - \frac{\eta}{4L} \right)$$

$$\overset{(a)}{\geq} \frac{1}{8L} \cdot \frac{1}{1 + \exp(\frac{\eta}{4L})} \left[ \eta - \frac{3\eta^2}{4} \right] + \frac{1}{8L} \frac{\eta}{2L} \langle u_1, E_2^w \rangle$$

$$+ \frac{1}{8L} \frac{1}{1 + \exp(-\frac{\eta}{4L})} \cdot \frac{\eta}{4L}$$

$$\overset{(b)}{\geq} \Omega\left(\frac{\eta}{L}\right),$$

where $(a)$ is due to the Lipchitz continuity of $f(x) = 1/(1 + e^x)$, and $(b)$ is due to $\langle u_1, E_2^w \rangle \leq -\eta^2/16$.

Similarly, for negative samples, since, we only flip the label $y_n$ from 1 to -1, we directly have

$$\lambda\left\langle E_1^{-w} - E_2^w, (-\nabla_W \mathcal{L}_1)E_L^w \right\rangle \leq -\Omega\left(\frac{\eta}{L}\right).$$

Following the same argument and algebra, we have

$$\lambda\left\langle E_1^w - E_2^w, (-\nabla_W \mathcal{L}_2)E_L^w \right\rangle \geq \Omega\left(\frac{\eta}{L}\right)$$

$$\lambda\left\langle E_1^{-w} - E_2^w, (-\nabla_W \mathcal{L}_2)E_L^w \right\rangle \leq -\Omega\left(\frac{\eta}{L}\right),$$

which implies that

$$\left\langle E_1^w - E_2^w, W_3 E_L^w \right\rangle \geq \Omega\left(\frac{\eta^2}{L}\right)$$

$$\left\langle E_1^{-w} - E_2^w, W_3 E_L^w \right\rangle \leq -\Omega\left(\frac{\eta^2}{L}\right),$$

Since $\langle u_1, E_\ell^w \rangle = 0$ for all $\ell \geq 2$, we have

$$\lambda\left\langle E_\ell^w - E_{\ell'}^w, (-\nabla_W \mathcal{L}_1)E_L^w \right\rangle = 0.$$

Finally, we aim to show that at time step $t = 3$, the attention layer also distinguishes between non-leading tokens. This can be done by noting that $\langle u_2, E_2^w - E_\ell^2 \rangle \leq -\Omega(\eta^2)$. Specifically, we have

$$\begin{aligned}
&\eta\lambda\langle E_2^w - E_\ell^w, (-\nabla_W \mathcal{L}_2)E_L^w \rangle \\
&= \frac{\eta}{2^L} \sum_{n \in I_L, \mathtt{C}(x_2^{(n)}) = \mathtt{C}(x_L^{(n)}) = w} (-J'_{(n,2)})y_n \varphi_2^{(n,2)}\left(\langle u_2, E_2^w \rangle - \mathtt{T}_2(X^{(n)})\right) \\
&\quad - \frac{\eta}{2^L} \sum_{n \in I_L, \mathtt{C}(x_\ell^{(n)}) = \mathtt{C}(x_L^{(n)}) = w} (-J'_{(n,2)})y_n \varphi_\ell^{(n,2)}\left(\langle u_2, E_\ell^w \rangle - \mathtt{T}_2(X^{(n)})\right) \\
&\overset{(a)}{=} \frac{\eta}{8}(-J'_{(+,t)})\varphi_2^{(+,2)}\left[\langle u_2, E_2^w \rangle - \langle u_2, E_\ell^w \rangle\right] \\
&\quad - \frac{\eta}{8}(-J'_{(-,t)})\varphi_2^{(-,2)}\left[\langle u_2, E_2^w \rangle - \langle u_2, E_\ell^w \rangle\right] \\
&\geq \Omega(\eta^4/L),
\end{aligned}$$

where $(a)$ is due to the fact that $\varphi_\ell^{(n,2)}$ are equal for any $\ell \geq 2$, and the last inequality follows from that $\langle u_2, E_2^w \rangle - \langle u_2, E_\ell^w \rangle \leq -\Omega(\eta^2)$, and $(-J'_{(+,2)} - (-J'_{(-,2)})) \leq -\Omega(\eta/L)$.

Thus, the proof is complete. □

Then we prove Theorem 4.1 through induction. Note that the statement in the following is essentially the same as that in Theorem 4.1 due to Lemma B.1.

**Theorem B.3** (Restatement of Theorem 4.1). *Choose* $\lambda = \Omega(L_{\max}^2), t_0 = O(1/(\eta L_{\max}))$, *and* $\eta = O(\min\{1/L_{\max}, 1/\lambda^{2/3}\})$. *we have*

*(1) The dynamics of linear layer $u$ is governed by the following inequalities.*

$$\langle u_t, E_1^w \rangle = \Theta(\eta t), \qquad \forall w \in \{a, b\},$$

$$\langle u_t, E_\ell^w \rangle = -\Theta(\eta^2 t^2), \qquad \forall \ell \geq 2, \forall w \in \{a, b\},$$

$$\langle u_t, E_2^w - E_\ell^w \rangle \leq -\Omega(\eta^2 t^2), \quad \forall w \in \{a, b\}$$

*(2) The dynamics of attention layer $W$ is governed by the following inequalities. For any length $L \leq L_{\max}$, we have*

$$\langle E_1^w - E_\ell^w, W_t E_L^w \rangle \geq \Omega(\eta^2 t^2 / L), \forall \ell \geq 2,$$

$$\langle E_2^w - E_1^{-w}, W_t E_L^w \rangle \geq \Omega(\eta^2 t^2 / L), \forall \ell \geq 2,$$

$$\langle E_2^w - E_\ell^w, W_t E_L^w \rangle \geq \Omega\left( \left(1 + \frac{\eta^3 t^2}{\lambda L^3}\right)^t \frac{\eta^4}{\lambda L} \right), \quad \forall \ell \geq 3,$$

*Proof.* By Lemma B.2, we know that the results hold true for $t = 3$. In the following, suppose the results are true for $t$. We will intensively use the following fact that characterize the norm of attention parameter $W_t$. Since

$$\|\nabla_W \mathcal{L}_t\| \leq 2 \sum_{L=2}^{L_{\max}} \frac{1}{2^L} \sum_{n \in I_L} \left| \sum_{\ell=1}^{L} \langle u_t, E_\ell^w \rangle \varphi_\ell^{(n,t)} \right|$$

$$\leq O(L_{\max} \eta t),$$

we have

$$\|W_t\| \leq \sum_{s=0}^{t-1} \eta \|\nabla_W \mathcal{L}_s\| \leq O(L_{\max} \eta^2 t^2).$$

**We first show that the dynamics of each token score $\langle u_t, E_\ell^w \rangle$ is true.**

For the first token, we have

$$\langle -\nabla_u \mathcal{L}_t, E_1^w \rangle = \sum_{L=1}^{L_{\max}} \frac{1}{2^L} \sum_{n \in I_L, \mathtt{C}(x_\ell^{(n)})=w} (-J_{(n,t)}')y_n \varphi_1^{(n,t)}$$

$$= \frac{1}{2} \cdot \frac{1}{1 + \exp\left(\langle u_t, E_1^w \rangle\right)}$$

$$+ \sum_{L=2}^{L_{\max}} \frac{1}{2^L} \sum_{n \in I_L, \mathtt{C}(x_1^{(n)})=w} (-J_{(n,0)}')y_n \varphi_1^{(n,0)}$$

$$+ \sum_{L=2}^{L_{\max}} \frac{1}{2^L} \sum_{n \in I_L, \mathtt{C}(x_1^{(n)})=w} \left(-J_{(n,t)}' + J_{(n,0)}'\right) y_n \varphi_1^{(n,0)}$$

$$+ \sum_{L=2}^{L_{\max}} \frac{1}{2^L} \sum_{n \in I_L, \mathtt{C}(x_1^{(n)})=w} (-J_{(n,t)}')y_n \left(\varphi_1^{(n,t)} - \varphi_1^{(n,0)}\right)$$

$$\overset{(a)}{=} \frac{1}{4} + \frac{1}{2} \left( \frac{1}{1 + \exp(\langle u_t, E_1^w \rangle)} - \frac{1}{2} \right)$$

$$+ \sum_{L=2}^{L_{\max}} \frac{1}{2^L} \sum_{n \in I_L, \mathtt{C}(x_1^{(n)})=w} \left(-J_{(n,t)}' + J_{(n,0)}'\right) y_n \varphi_1^{(n,0)} + \sum_{L=2}^{L_{\max}} \frac{1}{2^L} \sum_{n \in I_L} (-J_{(n,t)}')y_n \left(\varphi_1^{(n,t)} - \varphi_1^{(n,0)}\right),$$

where $(a)$ follows from the fact that $J_{(n,0)} = \frac{1}{2}$, and $\varphi_1^{(n,0)} = \frac{1}{L}$ if $n \in I_L$.

Then, by the Lipchitz continuity (Lemma A.1), we have

Hence,

$$
\left|\langle-\nabla_u\mathcal{L}_t, E_1^w\rangle - \frac{1}{4}\right| \leq \frac{1}{2}|\langle u_t, E_1^w\rangle| + \sum_{L=2}^{L_{\max}}\frac{1}{2^L}\sum_{n\in I_L, \mathtt{C}(x_1^{(n)})=w}\left|\frac{1}{2}-\frac{1}{1+\exp(y_n\mathrm{T}_t(X^{(n)}))}\right|\frac{1}{L} + \sum_{L=2}^{L_{\max}}\frac{1}{2}\cdot\frac{2\|W_t\|}{\lambda}
$$

$$
\overset{(a)}{\leq} \frac{1}{2}|\langle u_t, E_1^w\rangle| + \sum_{L=2}^{L_{\max}}\frac{1}{2^L}\sum_{n\in I_L, \mathtt{C}(x_1^{(n)})=w}\frac{1}{L}\left(\frac{1}{4}\left|\mathrm{T}_t(X^{(n)})\right| + \left|\mathrm{T}_t(X^{(n)})\right|^3\right) + \frac{L_{\max}\|W_t\|}{\lambda}
$$

$$
\overset{(b)}{\leq} \frac{1}{2}|\langle u_t, E_1^w\rangle| + \sum_{L=2}^{L_{\max}}\frac{1}{2L}\left(\max_{1\leq\ell\leq L}|\langle u_t, E_\ell^w\rangle|\right) + \frac{L_{\max}\|W_t\|}{\lambda}
$$

$$
\leq \frac{1}{2}|\langle u_t, E_1^w\rangle|(1+\log L_{\max}) + \frac{L_{\max}^2\eta^2 t^2}{\lambda}
$$

$$
\leq O(\eta t),
$$

where $(a)$ is due to Equation (2), $(b)$ follows from $\eta t = O(1)$, and the last inequality follows from $\lambda = \Omega(L_{\max}^2)$, and induction hypothesis.

Similarly, for $\ell \geq 2$, we have

$$
\langle-\nabla_u\mathcal{L}_t, E_\ell^w\rangle = \sum_{L=\ell}^{L_{\max}}\frac{1}{2^L}\sum_{n\in I_L, \mathtt{C}(x_\ell^{(n)})=w}(-J'_{(n,t)})y_n\varphi_\ell^{(n,t)}
$$

$$
= \sum_{L=\ell}^{L_{\max}}\frac{1}{2^L}\left(\sum_{n\in I_L^+, \mathtt{C}(x_\ell^{(n)})=w}(-J'_{(n,t)})\varphi_\ell^{(n,t)} - \sum_{n\in I_L^-, \mathtt{C}(x_\ell^{(n)})=w}(-J'_{(n,t)})\varphi_\ell^{(n,t)}\right)
$$

$$
= \sum_{L=\ell}^{L_{\max}}\frac{1}{4}\left((-J'_{(+,t)})\varphi_\ell^{(+,t)} - (-J'_{(-,t)})\varphi_\ell^{(-,t)}\right)
$$

$$
\overset{(a)}{\leq} \sum_{L=\ell}^{L_{\max}}\frac{1}{4}\left(\frac{1}{2}-\frac{1}{4}\mathrm{T}_t(X_L^{(+)}) - \frac{1}{2} - \frac{1}{4}\mathrm{T}_t(X_L^{(-)}) + \Theta(|\max_\ell\langle u_t, E_\ell^w\rangle|^3)\right)\varphi_\ell^{(-,t)}
$$

$$
\overset{(b)}{\leq} \sum_{L=\ell}^{L_{\max}}-\frac{1}{16}\left(\sum_{\ell'=1}^{L}(\varphi_{\ell'}^{(+,t)}+\varphi_{\ell'}^{(-,t)})\langle u_t, E_{\ell'}^w\rangle\right)\varphi_\ell^{(-,t)}
$$

$$
\overset{(c)}{\leq} -\Omega\left(\sum_{L=\ell}^{L_{\max}}\frac{1}{L}\left(\frac{\eta t}{L} - \frac{1}{L}\sum_{\ell'=2}^{L}\eta^2 t^2\right)\right)
$$

$$
= -\Omega(\eta t),
$$

where $(a)$ is due to Equation (2), $(b)$ follows from $\eta t = O(1)$, and $(c)$ is due to the induction hypothesis.

In addition, for any $\ell \geq 3$, we have

$$
\langle-\nabla_u\mathcal{L}_t, E_2^w - E_\ell^w\rangle = \sum_{L=2}^{\ell-1}\frac{1}{4}\left((-J'_{+,t})\varphi_2^{(+,t)} - (-J'_{-,t})\varphi_2^{(-,t)}\right)
$$

$$
+ \sum_{L=\ell}^{L_{\max}}\left((-J'_{(+,t)})(\varphi_2^{(+,t)} - \varphi_\ell^{(+,t)}) - (-J'_{(-,t)})(\varphi_2^{(-,t)} - \varphi_\ell^{(-,t)})\right)
$$

$$
\overset{(a)}{\leq} -\Omega(\eta t)
$$

$$
+ \sum_{L=\ell}^{L_{\max}}\left(\left(\frac{1}{2}-\frac{1}{4}\mathrm{T}_t(X^+)\right) + O(\eta^3 t^3)\right)\left(\varphi_2^{(+,t)} - \varphi_\ell^{(+,t)}\right)
$$

$$- \left( \frac{1}{2} + \frac{1}{4} \mathtt{T}_t(X^-) + O(\eta^3 t^3) \right) \left( \varphi_2^{(-,t)} - \varphi_\ell^{(-,t)} \right) \right)$$

$$\overset{(b)}{\leq} -\Omega(\eta t) + \sum_{L=\ell}^{L_{\max}} \frac{\|W_t\| \eta t}{\lambda}$$

$$\leq -\Omega(\eta t),$$

where $(a)$ follows from the same argument of the previous analysis on $\langle -\nabla_u \mathcal{L}_t, E_\ell^w \rangle$, and Equation (2), $(b)$) is due to the fact that $\mathtt{T}_t(X^{(n)}) \geq 0$, $\varphi_2^{(n,t)} - \varphi_\ell^{(n,t)} \geq 0$, and the last inequality follows from $\lambda = \Omega(L_{\max}^2)$.

Therefore,

$$\left| \langle u_{t+1}, E_1^w \rangle - \frac{\eta(t+1)}{4} \right| = \left| \sum_{s=2}^{t} \eta \langle -\nabla_u \mathcal{L}_s, E_1^w \rangle - \frac{\eta}{4} \right|$$

$$\leq O\left( \eta^2 t^2 \right)$$

Similarly, for $\ell \geq 2$,

$$\langle u_{t+1}, E_2^w \rangle = \sum_{s=2}^{t} \eta \langle -\nabla_u \mathcal{L}_s, E_2^w \rangle$$

$$\leq -\Omega(\eta^2 t^2).$$

In addition,

$$\langle u_{t+1}, E_2^w - E_\ell^w \rangle = \sum_{s=2}^{t} \eta \langle -\nabla_u \mathcal{L}_s, E_2^w - E_\ell^w \rangle$$

$$\leq -\Omega(\eta^2 t^2).$$

**We conclude that the dynamics of $u_t$ in Phase 1 is true.**

**Then, we show that the dynamics of $W_t$ in Phase 1 is true.**

In the following, we fix the length $L$, and focus on the $L$-th column of $W_t$.

We denote

$$A_{12}^{(t)} = \langle E_1^w - E_2^w, W_t E_L^w \rangle$$

$$B_{\ell 1}^{(t)} = \langle E_\ell^w - E_1^{-w}, W_t E_L^w \rangle$$

$$A_{2\ell}^{(t)} = \langle E_2^w - E_\ell^w, W_t E_L^w \rangle,$$

which are three quantities we aim to analyze.

Due to the gradient update in phase 1, We have

$$A_{12}^{(t+1)} - A_{12}^{(t)} = \eta \lambda \langle E_1^w - E_2^w, (-\nabla_W \mathcal{L}_t) E_L^w \rangle$$

$$= \frac{\eta}{2^L} \sum_{n \in I_L^+} (-J'_{(n,t)}) \varphi_1^{(n,t)} \left( \langle u_t, E_1^w \rangle - \mathtt{T}(X^{(n)}) \right)$$

$$- \frac{\eta}{2^L} \sum_{n \in I_L, \mathtt{C}(x_2^{(n)})=w} (-J'_{(n,t)}) y_n \varphi_2^{(n,t)} \left( \langle u_t, E_\ell^w \rangle - \mathtt{T}(X^{(n)}) \right)$$

$$= \frac{\eta}{8} (-J'_{(+,t)}) \left[ \varphi_1^{(+,t)} \left( \langle u_t, E_1^w \rangle - \mathtt{T}(X^{(+)}) \right) - \varphi_2^{(+,t)} \left( \langle u_t, E_2^w \rangle - \mathtt{T}(X^{(+)}) \right) \right]$$

$$+ \frac{\eta}{8}(-J'_{(+,t)})\varphi_1^{(+,t)}\left(\langle u_t, E_1^w\rangle - \mathtt{T}(X^{(+)})\right) + \frac{1}{8}(-J'_{(-,t)})\varphi_2^{(-,t)}\left(\langle u_t, E_2^w\rangle - \mathtt{T}(X^{(-)})\right)$$

$$\overset{(a)}{=} \frac{\eta}{8}\left(\frac{1}{2} - \frac{1}{4}\mathtt{T}_t(X^{(+)}) + O(|\mathtt{T}_t(X^{(+)})|^3)\right)\left[2\varphi_1^{(+,t)}\langle u_t, E_1^w\rangle - \varphi_2^{(+,t)}\langle u_t, E_2^w\rangle - (2\varphi_1^{(+,t)} - \varphi_2^{(+,t)})\mathtt{T}(X^{(+)})\right]$$

$$- \frac{\eta}{8}\left(\frac{1}{2} + \frac{1}{4}\mathtt{T}(X^{(-)}) + O(|\mathtt{T}_t(X^{(-)})|^3)\right)\varphi_2^{(-,t)}\left(|\langle u_t, E_2^w\rangle| + |\mathtt{T}(X^{(-)})|\right)$$

$$\overset{(b)}{\geq} \eta\Omega\left(\frac{2}{L}\eta t - \frac{1}{L}\eta^2 t^2 - \frac{1}{L}\eta t\right) - \eta O\left(\frac{1}{L}\eta^2 t^2 + \frac{1}{L^2}\eta t\right)$$

$$\geq \Omega\left(\frac{\eta^2 t}{L}\right),$$

where $(a)$ is due to Equation (2), and $(b)$ follows from the induction hypothesis. Thus,

$$A_{12}^{(t+1)} \geq \sum_{s=2}^{t}\left(A_{12}^{(s+1)} - A_{12}^{(s)}\right) + A_{12}^{(2)} = \Omega\left(\frac{\eta^2 t^2}{L}\right).$$

Similarly, for any $\ell \geq 2$, we have

$$B_{\ell 1}^{(t+1)} - B_{\ell 1}^{(t)} = \eta\lambda\langle E_\ell^w - E_1^{-w}, (-\nabla_W \mathcal{L}_t)E_L^w\rangle$$

$$= \frac{\eta}{2L}\sum_{n\in I_L, \mathtt{C}(x_\ell^{(n)})=\mathtt{C}(x_L^{(n)})=w}(-J'_{(n,t)})y_n\varphi_\ell^{(n,t)}\left(\langle u_t, E_\ell^w\rangle - \mathtt{T}(X^{(n)})\right)$$

$$+ \frac{\eta}{4}(-J_{(-,t)})\varphi_1^{(-,t)}\left(\langle u_t, E_1^w\rangle - \mathtt{T}(X^{(-)})\right)$$

$$= \frac{\eta}{8}(-J'_{(-,t)})\left[2\varphi_1^{(-,t)}\langle u_t, E_1^w\rangle - \varphi_\ell^{(-,t)}\langle u_t, E_\ell^w\rangle - (2\varphi_1^{(-,t)} - \varphi_\ell^{(-,t)})\mathtt{T}(X^{(-)})\right]$$

$$+ \frac{\eta}{8}(-J_{(+,t)})\varphi_\ell^{(+,t)}\left(\langle u_t, E_\ell^w\rangle - \mathtt{T}(X^{(+)})\right)$$

$$\overset{(a)}{\geq} \eta\Omega\left(\frac{2}{L}\eta t - \frac{1}{L}\eta^2 t^2 - \frac{1}{L}\eta t\right) - \eta O\left(\frac{1}{L}\eta^2 t^2 + \frac{1}{L^2}\eta t\right)$$

$$\geq \Omega\left(\frac{\eta^2 t}{L}\right),$$

where $(a)$ is due to Equation (2), and the induction hypothesis. Thus,

$$B_{\ell 1}^{(t+1)} \geq \Omega\left(\frac{\eta^2 t^2}{L}\right).$$

Finally, we show that the attention score at the second position surpasses those of other no-leading tokens.

We have

$$A_{2\ell}^{(t+1)} - A_{2\ell}^{(t)} = \eta\lambda\langle E_2^w - E_\ell^w, (-\nabla_W\mathcal{L}_t)E_L^w\rangle$$

$$= \frac{\eta}{2L}\sum_{n\in I_L, \mathtt{C}(x_2^{(n)})=\mathtt{C}(x_L^{(n)})=w}(-J'_{(n,t)})y_n\varphi_2^{(n,t)}\left(\langle u_t, E_2^w\rangle - \mathtt{T}(X^{(n)})\right)$$

$$- \frac{\eta}{2L}\sum_{n\in I_L, \mathtt{C}(x_\ell^{(n)})=\mathtt{C}(x_L^{(n)})=w}(-J'_{(n,t)})y_n\varphi_\ell^{(n,t)}\left(\langle u_t, E_\ell^w\rangle - \mathtt{T}(X^{(n)})\right)$$

$$= \frac{\eta}{8}(-J'_{(+,t)})\left[\varphi_2^{(+,t)}\langle u_t, E_2^w\rangle - \varphi_\ell^{(+,t)}\langle u_t, E_\ell^w\rangle - (\varphi_2^{(+,t)} - \varphi_\ell^{(+,t)})\mathtt{T}(X^+)\right]$$

$$- \frac{\eta}{8}(-J'_{(-,t)})\left[\varphi_2^{(-,t)}\langle u_t, E_2^w\rangle - \varphi_\ell^{(-,t)}\langle u_t, E_\ell^w\rangle - (\varphi_2^{(-,t)} - \varphi_\ell^{(-,t)})\mathtt{T}(X^-)\right]$$

$$= -\frac{\eta}{8}(-J'_{(+,t)})\left[\varphi_2^{(+,t)}\langle -u_t, E_2^w\rangle - \varphi_\ell^{(+,t)}\langle -u_t, E_\ell^w\rangle + (\varphi_2^{(+,t)} - \varphi_\ell^{(+,t)})\mathtt{T}(X^+)\right]$$

$$+ \frac{\eta}{8}(-J'_{(-,t)}) \left[ \varphi_2^{(-,t)} \langle -u_t, E_2^w \rangle - \varphi_\ell^{(-,t)} \langle -u_t, E_\ell^w \rangle + (\varphi_2^{(-,t)} - \varphi_\ell^{(-,t)}) \mathsf{T}(X^-) \right]$$

$$\overset{(a)}{=} -\frac{\eta}{8}(-J'_{(+,t)}) \varphi_\ell^{(+,t)} \left[ \exp(A_{2\ell}^{(t)}/\lambda) \langle -u_t, E_2^w \rangle - \langle -u_t, E_\ell^w \rangle + (\exp(A_{2\ell}^{(t)}/\lambda) - 1) \mathsf{T}(X^+) \right]$$

$$+ \frac{\eta}{8}(-J'_{(-,t)}) \varphi_\ell^{(-,t)} \left[ \exp(A_{2\ell}^{(t)}/\lambda) \langle -u_t, E_2^w \rangle - \langle -u_t, E_\ell^w \rangle + (\exp(A_{2\ell}^{(t)}/\lambda) - 1) \mathsf{T}(X^+) \right]$$

$$+ \frac{\eta}{8}(-J'_{(-,t)}) \varphi_\ell^{(-,t)} \left( \exp(A_{2\ell}^{(t)}/\lambda) - 1 \right) (\mathsf{T}(X^-) - \mathsf{T}(X^+))$$

$$= \frac{\eta}{8} \left( (-J'_{(-,t)}) \varphi_\ell^{(-,t)} - (-J'_{(+,t)}) \varphi_\ell^{(+,t)} \right)$$

$$\times \left[ \langle -u_t, E_2^w \rangle - \langle -u_t, E_\ell^w \rangle + (\exp(A_{2\ell}^{(t)}/\lambda) - 1)(\langle -u_t, E_2^w \rangle + \mathsf{T}(X^+)) \right]$$

$$+ \frac{\eta}{8}(-J'_{(-,t)}) \varphi_\ell^{(-,t)} \left( \exp(A_{2\ell}^{(t)}/\lambda) - 1 \right) (\mathsf{T}(X^-) - \mathsf{T}(X^+))$$

$$\overset{(b)}{\geq} \eta\Omega \left( \frac{\eta t}{L} \left( \exp(A_{2\ell}^{(t)}/\lambda) - 1 \right) \frac{\eta t}{L} \right) - \eta O \left( \frac{1}{L} \left( \exp(A_{2\ell}^{(t)}/\lambda) - 1 \right) \frac{L_{\max}\eta^2 t^2}{\lambda} \eta t \right)$$

$$\overset{(c)}{\geq} \Omega \left( \frac{\eta^3 t^2}{L} (\frac{1}{L^2} - \frac{L_{\max}\eta t}{L\lambda}) \right) \frac{A_{2\ell}^{(t)}}{\lambda},$$

where $(a)$ follows from the definition of softmax, $(b)$ is due to the induction hypothesis, and $(c)$ is due to $\lambda = \Omega(L_{\max})$. By noting that $A_{2\ell}^{(3)} \geq \eta^4/L$, we have

$$A_{2\ell}^{(t)} \geq \Omega \left( \left( 1 + \frac{\eta^3 t^2}{\lambda L^3} \right)^t \frac{\eta^4}{\lambda L} \right).$$

**We conclude that the dynamics of $W_t$ in Phase 1 is true.**

$\square$

### B.2. Phase 2

In this section, we analyze the training dynamics during Phase 2. Roughly speaking, both the linear layer's parameter vector $\mathbf{u}_t$ and the attention layer's parameters increase in norm over time. However, due to the scaling factor $\lambda$, the linear layer dominates the loss reduction, contributing more significantly to optimization progress than the attention layer. In the following, we leverage implicit bias theory to demonstrate that the growth of $\|\mathbf{u}_t\|$ induces a sublinear convergence rate for the loss, governed by $\mathcal{O}(1/t)$. On the other hand, the attention layer does not change significantly.

First, we show that at the end of phase 1 $(t = t_0)$, the attention layer make data samples separable.

**Proposition B.4** (Restatement of Proposition 4.3). *Let $v^{(n)} = \sum_{\ell=1}^{L} x_\ell^{(n)} \varphi_\ell^{(n,t_0)}$. Then, at the end of phase 1, the dataset $\{(v^{(n)}, y_n)\}$ is separable.*

*Proof.* Let $u = \sum_{w \in \{a, b\}} (E_1^w - E_2^w)$. Then, for any positive sequence, where $y_n = 1$, we have

$$\langle u, y_n \sum_{\ell=1}^{L} x_\ell^{(n)} \varphi_\ell^{(n,t_0)} \rangle = \varphi_1^{(n,t_0)} - \varphi_2^{(n,t_0)}$$

$$= \varphi_2^{(n,t_0)} \left( \exp\left( \langle E_1^w - E_2^w, W_t E_L^w \rangle \right) - 1 \right)$$

$$> 0,$$

where the last inequality is due to Theorem 4.1.

Similarly, for any negative samples, we have

$$\langle u, y_n \sum_{\ell=1}^{L} x_\ell^{(n)} \varphi_\ell^{(n,t_0)} \rangle = -\varphi_1^{(n,t_0)} + \varphi_2^{(n,t_0)}$$

$$= \varphi_1^{(n,t_0)} \left( \exp\left( \langle E_2^w - E_1^{-w}, W_t E_L^w \rangle \right) - 1 \right)$$

$$> 0,$$

where the last inequality is due to Theorem 4.1.

Thus, The proof is complete. $\qquad\square$

**Parameter setup.** Before we present the technical lemmas, we first introduce two parameters. Let

$$\begin{cases} T = \min\left\{ \dfrac{\lambda}{\sqrt{3}\|u_{EP}^*\|\eta L_{\max}}, \dfrac{\lambda^{2/3}}{2^{2/3}\eta L_{\max}} \right\} = \dfrac{\lambda^{2/3}}{2^{2/3}\eta L_{\max}} \\ C_0 = 2\log 4L_{\max} + 2 \end{cases}$$

We remark that since $\lambda = \Omega(L_{\max}^2)$, $T$ is well defined. We first provide the property of $T$. Note that for any $t$, we have

$$\|u_t\| \le \|u_t - u_{t_2}\| + \|u_{t_2}\|$$
$$\le \eta \sum_{s=0}^{t} \|\nabla_u \mathcal{L}_t\|$$
$$\le \eta L_{\max} t$$

Similarly,

$$\|W_t - W_{t_0}\| \le \sum_{s=t_0}^{t} \frac{\eta}{\lambda} \|\nabla_W \mathcal{L}_t\|$$
$$\le \frac{\eta}{\lambda} \sum_{s=t_0}^{t} \|u_s\| L_{\max}$$
$$\le \frac{\eta^2 L_{\max}^2 t^2}{\lambda}$$

Therefore, for any $t \le T$, we have

$$\|u_{EP}^*\|\|W_t - W_{t_0}\| \le \frac{\lambda}{3} \text{ and } \|u_t\|\|W_t - W_{t_0}\| \le \frac{\lambda}{2}.$$

Next, we provide the key lemma in phase 2, which characterizes the alignment of the gradient $-\nabla_u \mathcal{L}_t$ and the logarithm growth of $\|u_t\|$.

**Lemma B.5.** *Let $u_{EP}^*$ be the solution of the following problem.*

$$u_{EP}^* = \arg\min \|u\|, \quad \left\langle u, y_n \sum_{\ell=1}^{L} \varphi_\ell^{(n,t_0)} x_\ell^{(n)} \right\rangle \ge 1, \quad \forall n \in I_L, L \ge 1.$$

*Then, for all $t_0 < t \le T$, we have*

$$\left\langle -\nabla_u \mathcal{L}_t, \frac{u_t}{\|u_t\|} \right\rangle \le \left( 1 + \frac{C_0\|u_{EP}^*\|}{\|u_t\|} \right) \left\langle -\nabla_u \mathcal{L}_t, \frac{u_{EP}^*}{\|u_{EP}^*\|} \right\rangle,$$

*and*

$$\|u_t\| \ge \frac{3\|u_{EP}^*\|}{5} \log(t - t_0 - 1) - \frac{3\|u_{EP}^*\|}{5} \log \frac{9\|u_{EP}^*\|^2}{\eta}.$$

*Proof.* First, note that for any $t \le T$, we have the following lower bound of $C_0$.

$$C_0 \ge \frac{\lambda \log 4L_{\max} + 2\|u_t\|\|W_t - W_{t_0}\|}{\lambda - \|u_{EP}^*\|\|W_t - W_{t_0}\|}.$$

**We aim to show that** $\mathcal{L}\left(u_{EP}^*\left(\frac{\|u_t\|}{\|u_{EP}^*\|} + C_0\right), W_t\right) \leq \mathcal{L}(u_t, W_t).$

Since $u_t$ is not the optimal solution of the problem

$$\arg\min \|u\|, \quad \left\langle u, y_n \sum_{\ell=1}^{L} \varphi_\ell^{(n,t_0)} x_\ell^{(n)} \right\rangle \geq 1, \quad \forall n \in I_L, L \geq 1,$$

we have at least one sample $n$ of length $L$ such that

$$\left\langle \frac{u_t}{\|u_t\|}, y_n \sum_{\ell=1}^{L} \varphi_\ell^{(n,t_0)} x_\ell^{(n)} \right\rangle < \frac{1}{\|u_{EP}^*\|},$$

which implies there are at least $2^{L-1}$ samples satisfies the same inequality.

Hence, for this $n$, we have

$$y_n T_t(X^{(n)}) = \left\langle u_t, y_n \sum_{\ell=1}^{L} \varphi_\ell^{(n,t)} x_\ell^{(n)} \right\rangle$$

$$= \left\langle u_t, y_n \sum_{\ell=1}^{L} \varphi_\ell^{(n,t_0)} x_\ell^{(n)} \right\rangle + \left\langle u_t, y_n \sum_{\ell=1}^{L} (\varphi_\ell^{(n,t)} - \varphi_\ell^{(n,t_0)}) x_\ell^{(n)} \right\rangle$$

$$\leq \frac{\|u_t\|}{\|u_{EP}^*\|} + \|u_t\| \frac{\|W_t - W_{t_0}\|}{\lambda},$$

where the inequality is due to the Cauchy's inequality and the Lipchitz continuity of softmax function (Lemma A.1).

Thus,

$$\mathcal{L}(u_t, W_t) = \sum_{L=1}^{L_{\max}} \frac{1}{2^L} \sum_{n \in I_L} \log\left(1 + \exp\left(-y_n T_t(X^{(n)})\right)\right)$$

$$\geq \frac{1}{2} \log\left(1 + \exp\left(-\frac{\|u_t\|}{\|u_{EP}^*\|} - \frac{\|u_t\|\|W_t - W_{t_0}\|}{\lambda}\right)\right)$$

$$\geq \frac{1}{4} \exp\left(-\frac{\|u_t\|}{\|u_{EP}^*\|} - \frac{\|u_t\|\|W_t - W_{t_0}\|}{\lambda}\right),$$

where the last inequality follows from $\log(1+x) > \frac{x}{2}$ when $x \in (0,1)$.

On the other hand, we have

$$\left\langle u_{EP}^*, y_n \sum_{\ell=1}^{L} \varphi_\ell^{(n,t)} x_\ell^{(n)} \right\rangle = \left\langle u_{EP}^*, y_n \sum_{\ell=1}^{L} \varphi_\ell^{(n,t_0)} x_\ell^{(n)} \right\rangle + \left\langle u_{EP}^*, y_n \sum_{\ell=1}^{L} (\varphi_\ell^{(n,t)} - \varphi_\ell^{(n,t_0)}) x_\ell^{(n)} \right\rangle$$

$$\geq 1 - \|u_{t_0}\|^* \frac{\|W_t - W_{t_0}\|}{\lambda},$$

where the inequality is due to the definition of $u_{EP}^*$ and Cauchy's inequality.

Thus,

$$\mathcal{L}\left(u_{EP}^*\left(\frac{\|u_t\|}{\|u_{EP}^*\|} + C_0\right), W_t\right) \leq L_{\max} \log\left(1 + \exp\left(-\left(1 - \frac{\|u_{EP}^*\|\|W_t - W_{t_0}\|}{\lambda}\right)\left(\frac{\|u_t\|}{\|u_{EP}^*\|} + C_0\right)\right)\right)$$

$$\leq L_{\max} \exp\left(-\frac{\|u_t\|}{\|u_{EP}^*\|} + \frac{\|u_t\|\|W_t - W_{t_0}\|}{\lambda}\right) \exp\left(-\log 4L_{\max} - 2\frac{\|u_t\|\|W_t - W_{t_0}\|}{\lambda}\right)$$

$$= \frac{1}{4} \exp\left(-\frac{\|u_t\|}{\|u_{EP}^*\|} - \frac{\|u_t\|\|W_t - W_{t_0}\|}{\lambda}\right)$$

**Therefore, we conlucde that** $\mathcal{L}\left(u_{EP}^*\left(\frac{\|u_t\|}{\|u_{EP}^*\|} + C_0\right), W_t\right) \leq \mathcal{L}(u_t, W_t).$

Since $\mathcal{L}(u, W_t)$ is convex respect to $u$ for any $W_t$, by the first order optimality, we have

$$\left\langle \nabla_u \mathcal{L}_t, u_{EP}^*\left(\frac{\|u_t\|}{\|u_{EP}^*\|} + C_0\right) - u_t \right\rangle \leq 0$$

By rearranging, we conclude that

$$\left\langle -\nabla_u \mathcal{L}_t, \frac{u_t}{\|u_t\|} \right\rangle \leq \left(1 + \frac{C_0\|u_{EP}^*\|}{\|u_t\|}\right) \left\langle -\nabla_u \mathcal{L}_t, \frac{u_{EP}^*}{\|u_{EP}^*\|} \right\rangle.$$

.

**Next, we show that the norm of $u_t$ grows at least logarithmically.**

On one hand, we have

$$\left\langle -\nabla_u \mathcal{L}_s, \frac{u_{EP}^*}{\|u_{EP}^*\|} \right\rangle = \sum_{L=1}^{L_{\max}} \frac{1}{2^L} \sum_{n \in I_L} \left(-J'_{(n,s)}\right) \left\langle \frac{u_{t_0}}{\|u_{EP}^*\|}, \sum_{\ell=1}^L \varphi_\ell^{(n,s)} x_\ell^{(n)} \right\rangle$$

$$\geq \sum_{L=1}^{L_{\max}} \frac{1}{2^L} \sum_{n \in I_L} \frac{1}{1 + \exp(y_n \mathrm{T}_s(X^{(n)}))} \left(\frac{1}{\|u_{EP}^*\|} - \frac{\|W_s - W_{t_0}\|}{\lambda}\right)$$

$$\overset{(a)}{\geq} \frac{1}{2} \frac{1}{1 + \exp\left(\frac{\|u_s\|}{\|u_{EP}^*\|} + \|u_t\|\|W_s - W_{t_0}\|/\lambda\right)} \left(\frac{1}{\|u_{EP}^*\|} - \frac{\|W_s - W_{t_0}\|}{\lambda}\right)$$

$$\overset{(b)}{\geq} \frac{1}{3\|u_{EP}^*\|} \frac{1}{1 + 2\exp\left(\frac{\|u_s\|}{\|u_{EP}^*\|}\right)},$$

where $(a)$ follows from the non-optimality of $u_s$, and $(b)$ is due to that $\|W_s - W_{t_0}\|/\lambda \leq \frac{1}{2\|u_t\|}$

Thus,

$$\|u_s\| \geq \frac{\eta}{3\|u_{EP}^*\|} \sum_{s'=t_0}^{s-1} \frac{1}{1 + 2\exp\left(\frac{\|u_{s'}\|}{\|u_{EP}^*\|}\right)} - \|u_{t_0}\|$$

Let $t_k = \max_t\{t | \forall s \leq t, \|u_s\| \leq k\}$. We next show that $t_k$ has an upper bound.

We have for any $t \leq t_k$,

$$k \geq \|u_t\| \geq \frac{\eta}{3\|u_{EP}^*\|} \frac{(t - t_0)}{1 + 2\exp(\frac{k}{\|u_{EP}^*\|})} - \|u_{t_0}\|$$

Thus,

$$t - t_0 \leq \frac{3\|u_{EP}^*\|}{\eta}(k + \|u_{t_0}\|)\left(1 + 2\exp\left(\frac{k}{\|u_{EP}^*\|}\right)\right)$$

$$\leq \frac{9\|u_{EP}^*\|^2}{\eta}\left(\frac{k}{\|u_{EP}^*\|} + 1\right)\exp\left(\frac{k}{\|u_{EP}^*\|}\right)$$

$$\leq \frac{14\|u_{EP}^*\|^2}{\eta}\left(\frac{2k}{3\|u_{EP}^*\|} + 1\right)\exp\left(\frac{k}{\|u_{EP}^*\|}\right)$$

$$\leq \frac{14\|u_{EP}^*\|^2}{\eta}\exp\left(\frac{5k}{3\|u_{EP}^*\|}\right),$$

where the last inequality is due to $x + 1 \leq e^x$. Hence, $t_k$ has upper bound:

$$t_k - t_0 \leq \frac{14\|u_{EP}^*\|^2}{\eta}\exp\left(\frac{5k}{3\|u_{EP}^*\|}\right).$$

Therefore, by the definition of $t_k$, we have

$$\|u_{t_k+1}\| > k$$
$$\geq \frac{3\|u_{EP}^*\|}{5} \log \left( \frac{\eta}{14\|u_{EP}^*\|^2} (t_k - t_0) \right)$$

Equivalently, we have

$$\|u_t\| \geq \frac{3\|u_{EP}^*\|}{5} \log(t - t_0 - 1) - \frac{3\|u_{EP}^*\|}{5} \log \frac{14\|u_{EP}^*\|^2}{\eta}.$$

**We conclude that the norm of $u_t$ grows at least logarithmically.**

The proof is complete.

$\square$

In the following, we show that there exists a threshold $t_2$, such that after time step $t_2$, the transformer can correctly label the data, which eventually leads to loss decay.

**Lemma B.6** (Formal version of Theorem 4.4). *There exists $t_2$ such that for any $t \geq t_2$, we have*

$$\left\langle \frac{u_t}{\|u_t\|}, \frac{u_{EP}^*}{\|u_{EP}^*\|} \right\rangle \geq 1 - \frac{1}{2} \left( \frac{1}{6\|u_{EP}^*\|} - \frac{1}{\|u_t\|} \right)^2,$$

*and*

$$y_n \mathrm{T}_t(X^{(n)}) \geq \frac{5\|u_t\|}{6\|u_{EP}^*\|}. \tag{6}$$

*Proof.* The proof consists of two parts that leverage Lemma B.5 in a different way. In the first part, we aim to find an interval $[t_2, t_3]$ such that the result holds. In the second part, we aim to prove that for any $t \geq t_3$, the result holds.

Let

$$t_1 = t_0 + 1 + \frac{14\|u_{EP}^*\|^2}{\eta} \exp \left( \frac{5}{3\Delta\|u_{EP}^*\|} \right).$$

Then, we have

$$\|u_t\| \geq \frac{1}{\Delta},$$

where $\Delta$ will be determined later.

**We aim to find an interval $[t_2, t_3]$ such that Equation (6) is true.**

By Lemma B.5, for $t \geq t_1$, we have

$$\left\langle -\nabla_u \mathcal{L}_t, \frac{u_t}{\|u_t\|} \right\rangle \leq \left( 1 + \frac{C_0\|u_{EP}^*\|}{\|u_t\|} \right) \left\langle -\nabla_u \mathcal{L}_t, \frac{u_{EP}^*}{\|u_{EP}^*\|} \right\rangle$$
$$\leq (1 + \Delta C_0\|u_{EP}^*\|) \left\langle -\nabla_u \mathcal{L}_t, \frac{u_{EP}^*}{\|u_{EP}^*\|} \right\rangle,$$

which implies

$$\left\langle u_{t+1} - u_t, \frac{u_{EP}^*}{\|u_{EP}^*\|} \right\rangle \geq \frac{1}{1 + \Delta C_0\|u_{EP}^*\|} \left\langle u_{t+1} - u_t, \frac{u_t}{\|u_t\|} \right\rangle$$
$$\geq \frac{1}{1 + \Delta C_0\|u_{EP}^*\|} \left( \|u_{t+1}\| - \|u_t\| - \frac{\|u_{t+1} - u_t\|^2}{2\|u_t\|} \right)$$
$$\geq \frac{1}{1 + \Delta C_0\|u_{EP}^*\|} \left( \|u_{t+1}\| - \|u_t\| - \frac{\eta^2 L_{\max}^2 \Delta}{2} \right).$$

By telescoping from $t_1$ to $t - 1$ and rearranging, we have

$$\left\langle \frac{u_t}{\|u_t\|}, \frac{u_{EP}^*}{\|u_{EP}^*\|} \right\rangle \geq \frac{1}{1 + \Delta C_0 \|u_{EP}^*\|} \left( 1 - \frac{\|u_{t_1}\|}{\|u_t\|} - \frac{\eta^2 L_{\max}^2 \Delta(t - t_1)}{2\|u_t\|} \right) - \frac{\|u_{t_1}\|}{\|u_t\|}.$$

Now choose $t_2$ such that

$$t_2 = t_0 + 1 + \frac{9\|u_{EP}^*\|}{\eta} \exp\left( \frac{5\|u_{t_1}\|}{3\Delta\|u_{EP}^*\|} \right),$$

which gives us $\|u_t\| \geq \frac{\|u_{t_1}\|}{\Delta}$.

Thus,

$$\left\langle \frac{u_t}{\|u_t\|}, \frac{u_{EP}^*}{\|u_{EP}^*\|} \right\rangle \geq \frac{1}{1 + \Delta C_0 \|u_{EP}^*\|} \left( 1 - \frac{\|u_{t_1}\|}{\|u_t\|} - \frac{\eta^2 L_{\max}^2 \Delta(t - t_1)}{2\|u_t\|} \right) - \frac{\|u_{t_1}\|}{\|u_t\|}$$

$$\overset{(a)}{\geq} 1 - \frac{\Delta C_0 \|u_{EP}^*\|}{1 + \Delta C_0 \|u_{EP}^*\|} - 2\|u_{t_1}\|/\|u_t\| - \eta^2 L_{\max}^2 \Delta^3 (t - t_1)/2$$

$$\geq 1 - \frac{\Delta C_0 \|u_{EP}^*\|}{1 + \Delta C_0 \|u_{EP}^*\|} - 2\Delta - \eta^2 L_{\max}^2 \Delta^3 (t - t_1)/2,$$

where $(a)$ follows from $\|u_t\| \geq \|u_{t_1}\|/\Delta \geq 1/\Delta^2$, and the last inequality is due to that $\|u_{t_1}\|/\|u_t\| \leq \Delta$.

By choosing $\Delta = \Theta\left( \frac{1}{C_0 \|u_{EP}^*\|^3} \right)$, and noting that $t \leq T = \frac{\lambda^{2/3}}{\eta L_{\max}}$, we conclude that for $t \in [t_2, T]$, the following inequalities hold.

$$\left\| \frac{u_t}{\|u_t\|} - \frac{u_{EP}^*}{\|u_{EP}^*\|} \right\| = \sqrt{2 - 2\left\langle \frac{u_t}{\|u_t\|}, \frac{u_{EP}^*}{\|u_{EP}^*\|} \right\rangle}$$

$$\leq \sqrt{\frac{2\Delta C_0 \|u_{EP}^*\|}{1 + \Delta C_0 \|u_{EP}^*\|} + 4\Delta + \eta^2 L_{\max}^2 \Delta^2 (t - t_1)/2}$$

$$\leq \sqrt{\Delta} \sqrt{2 C_0 \|u_{EP}^*\| + 4 + \eta^2 L_{\max}^2 \Delta^2 (t_3 - t_1)/2}$$

$$\leq \frac{1}{6\|u_{EP}^*\|} - \Delta$$

$$\leq \frac{1}{6\|u_{EP}^*\|} - \frac{1}{\|u_t\|},$$

where the last inequality is due to that $\|u_t\| \geq 1/\Delta$.

Hence,

$$y_n \mathrm{T}_t(X^{(n)}) = \left\langle u_t, y_n \sum_{\ell=1}^{L} \varphi_\ell^{(n,t)} x_\ell^{(n)} \right\rangle$$

$$\overset{(a)}{\geq} \left\langle u_t, y_n \sum_{\ell=1}^{L} \varphi_\ell^{(n,t_0)} x_\ell^{(n)} \right\rangle - \frac{\|u_t\|\|W_t - W_{t_0}\|}{\lambda}$$

$$= \frac{\|u_t\|}{\|u_{EP}^*\|} \left\langle u_{EP}^*, y_n \sum_{\ell=1}^{L} \varphi_\ell^{(n,t_0)} x_\ell^{(n)} \right\rangle + \left\langle u_t - \|u_t\| \frac{u_{EP}^*}{\|u_{EP}^*\|}, y_n \sum_{\ell=1}^{L} \varphi_\ell^{(n,t_0)} x_\ell^{(n)} \right\rangle - \frac{\|u_t\|\|W_t - W_{t_0}\|}{\lambda}$$

$$\geq \frac{\|u_t\|}{\|u_{EP}^*\|} - \|u_t\| \left\| \frac{u_t}{\|u_t\|} - \frac{u_{t_0}}{\|u_{EP}^*\|} \right\| - \frac{\|u_t\|\|W_t - W_{t_0}\|}{\lambda}$$

$$\overset{(b)}{\geq} \frac{\|u_t\|}{\|u_{EP}^*\|} - \|u_t\| \left( \frac{1}{6\|u_{EP}^*\|} - \frac{1}{\|u_t\|} \right) - 1$$

$$= \frac{5\|u_t\|}{6\|u_{EP}^*\|},$$

where $(a)$ follows from Cauchy's inequality and the Lipchitz continuity of softmax function (Lemma A.1), and $(b)$ is the due that $t \leq T$.

The proof is complete.

$\square$

**Theorem B.7** (Restatement of Theorem 4.5). *Recall that* $T = \Theta\left(\frac{\lambda^{2/3}}{\eta L_{\max}}\right)$. *For any* $t \leq T$, *we have*

$$\mathcal{L}_t = O\left(\frac{L_{\max}\|u_{EP}^*\|^2}{\eta\sqrt{t}}\right)$$

*Proof.* By Lemmas B.5 and B.6, we have proved that for any $T \geq t \geq t_2$, we have

$$y_n \mathrm{T}_t(X^{(n)}) \geq \frac{5\|u_t\|}{6\|u_{EP}^*\|}, \quad \forall n.$$

and

$$\|u_t\| \geq \frac{3\|u_{EP}^*\|}{5} \log \frac{\eta(t - t_0 - 1)}{14\|u_{EP}^*\|^2}.$$

Thus,

$$\mathcal{L}_t = \sum_{L=1}^{L_{\max}} \frac{1}{2^L} \log\left(1 + \exp(-y_n \mathrm{T}_t(X^{(n)}))\right)$$

$$\leq L_{\max} \log\left(1 + \frac{14\|u_{EP}^*\|^2}{\eta} \frac{1}{\sqrt{t - t_0 - 1}}\right)$$

$$\leq O\left(\frac{L_{\max}\|u_{EP}^*\|^2}{\eta\sqrt{t}}\right).$$

Finally, since $T = \Theta\left(\frac{\lambda^{2/3}}{\eta L_{\max}}\right)$ we note that

$$\mathcal{L}_T \leq O\left(\frac{\eta^{1/2} L_{\max}^{2.5}\|u_{EP}^*\|^2}{\lambda^{1/3}}\right).$$

$\square$

## C. Proofs of Parity Check Problem

In this section, we provide the proof of training dynamics of transformers on parity check with CoT. Our strategy is similar to that used in Appendix B. Since once the data after the attention layer is separable, the analysis of the dynamics would be the same. Hence, we omit the phase 2 analysis in parity check and focus on showing that at the end of phase 1, the attention layer makes the data separable.

We first show that the token scores only depend on the position, which helps us to reduce the complexity of the subsequent proof.

**Lemma C.1.** *During the entire training process, for any $w \in \{a, b\}$, we have*

$$\langle u_t, E_\ell^w \rangle = \langle u_t, E_\ell^{-w} \rangle, \quad \forall \ell \geq 1.$$

*For attention scores, when $L < L_0$, we have the following equalities.*

$$\begin{cases} \langle E_1^w, W_t E_L^w \rangle = \langle E_1^{-w}, W_t E_L^{-w} \rangle. \\ \langle E_\ell^w, W_t E_L^w \rangle = \langle E_\ell^{-w}, W_t E_L^w \rangle, \quad \forall \ell \geq 2. \end{cases}$$

*When $L \geq L_0$, let $\ell_0 = L - L_0 + 1$, then we have the following equalities.*

$$\begin{cases} \left\langle E_{\ell_0}^w, W_t E_L^w \right\rangle = \left\langle E_{\ell_0}^{-w}, W_t E_L^{-w} \right\rangle. \\ \left\langle E_\ell^w, W_t E_L^w \right\rangle = \left\langle E_\ell^{-w}, W_t E_L^w \right\rangle, \quad \forall \ell \neq \ell_0. \end{cases}$$

*Proof.* We only check the last two equalities when $L \geq L_0$, since the others can proved similarly to Lemma B.1.

Note that the results are valid for $t = 0$. Assume the results hold at time $t$, we aim to prove the results hold for $t + 1$. It suffices to prove the following equalities.

$$\left\langle E_{\ell_0}^w, (-\nabla_W \mathcal{L}_t) E_L^w \right\rangle = \left\langle E_{\ell_0}^{-w}, (-\nabla_W \mathcal{L}_t) E_L^{-w} \right\rangle. \tag{7}$$

$$\left\langle E_\ell^w, (-\nabla_W \mathcal{L}_t) E_L^w \right\rangle = \left\langle E_\ell^{-w}, (-\nabla_W \mathcal{L}_t) E_L^w \right\rangle, \quad \forall \ell \neq \ell_0. \tag{8}$$

**We first show that Equation (8) is true.**

For $\ell \neq \ell_0 = L - L_0 + 1$, we have

$$\left\langle E_\ell^w, (\nabla_W \mathcal{L}_t) E_L^w \right\rangle = \frac{1}{2^L} \sum_{n \in I_L, \mathtt{C}(x_\ell^{(n)}) = \mathtt{C}(x_L^{(n)}) = w} J'_{(n,t)} y_n \varphi_\ell^{(n,t)} \left( u_t^\top x_\ell^{(n)} - \mathtt{T}_t(X_L^{(n)}) \right)$$

$$= \frac{1}{2^L} \sum_{n \in I_L, \mathtt{C}(x_\ell^{(n)}) = \mathtt{C}(x_L^{(n)}) = w} \frac{-1}{1 + \exp\left(y_n \mathtt{T}_t(X^{(n)})\right)} y_n \varphi_\ell^{(n,t)} \left( u_t^\top x_\ell^{(n)} - \mathtt{T}_t(X_L^{(n)}) \right)$$

For any $n \in I_L$ satisfying $\mathtt{C}(x_\ell^{(n)}) = \mathtt{C}(x_L^{(n)}) = w$, let $n'$ be the sample that only replace $w$ with $-w$ at the $\ell$-th position. Then, due to the induction hypothesis, we have $\varphi_\ell^{(n,t)} = \varphi_\ell^{(n',t)}$, and $\mathtt{T}_t(X_L^{(n)}) = \mathtt{T}_t(X_L^{(n')})$. Since $\ell \geq 2$, changing one token at the position $\ell$ does not change the label, we have $y_n = y_{n'}$. Therefore, we have

$$\left\langle E_\ell^w, (\nabla_W \mathcal{L}_t) E_L^w \right\rangle = \left\langle E_\ell^{-w}, (\nabla_W \mathcal{L}_t) E_L^w \right\rangle, \quad \forall \ell \geq 2.$$

**We conclude that Equation (8) is true.**

**Then, we show that Equation (7) is true.**

Note that

$$\left\langle E_{\ell_0}^w, (\nabla_W \mathcal{L}_t) E_L^w \right\rangle = \frac{1}{2^L} \sum_{n \in I_L, \mathtt{C}(x_{\ell_0}^{(n)}) = \mathtt{C}(x_L^{(n)}) = w} J'_{(n,t)} \varphi_1^{(n,t)} \left( u_t^\top x_1^{(n)} - \mathtt{T}_t(X_L^{(n)}) \right)$$

$$= \frac{1}{2^L} \sum_{n \in I_L, \mathtt{C}(x_{\ell_0}^{(n)}) = \mathtt{C}(x_L^{(n)}) = w} \frac{-1}{1 + \exp\left(y_n \mathtt{T}_t(X_L^{(n)})\right)} \varphi_1^{(n,t)} \left( u_t^\top x_1^{(n)} - \mathtt{T}_t(X_L^{(n)}) \right).$$

Now, for any $n \in I_L$ satisfying $\mathtt{C}(x_{\ell_0}^{(n)}) = \mathtt{C}(x_L^{(n)}) = w$, let $n'$ be the sample that flips the $\ell_0$-th and the last tokens at the same time, i.e., $\mathtt{C}(x_{\ell_0}^{(n')}) = \mathtt{C}(x_L^{(n')}) = -w$. Then, due to the induction hypothesis, we have $\varphi_\ell^{(n,t)} = \varphi_\ell^{(n',t)}$, $y_n = y_{n'}$ and $\mathtt{T}_t(X_L^{(n)}) = \mathtt{T}_t(X_L^{(n')})$. Therefore, we have

$$\left\langle E_{\ell_0}^w, (\nabla_W \mathcal{L}_t) E_L^w \right\rangle = \left\langle E_{\ell_0}^{-w}, (\nabla_W \mathcal{L}_t) E_L^{-w} \right\rangle.$$

**We conclude that Equation (7) is true.**

Therefore, the proof is complete by induction. $\qquad \square$

Due to above lemma, for each length $L$, we only need to analyze two types of sequence, i.e., the one with positive label and the one with negative label. We also use $X_L^{(+)} = [E_1^w, E_2^w, \ldots, E_L^w s]$ to represent the sequence with positive labels, and $X_L^{(-)} = [E_1^{-w}, E_2^w, \ldots, E_L^2]$ to represent the sequence with negative labels, similar to Appendix B.

Next, we characterize the initialization dynamics.

**Lemma C.2** (Initialization). *At the beginning ($t = 2, 3$), for the linear layer, we have*

$$\langle u_2, E_1^w \rangle = \Theta(\eta t), \quad \forall w,$$
$$\langle u_2, E_\ell^w \rangle = -\Theta(\eta^2 t^2), \quad \forall \ell \geq 2, \forall w,$$
$$\langle u_2, E_2^w - E_\ell^w \rangle \leq -\Omega(\eta^2 t^2), \quad \forall \ell \geq 3, \forall w.$$

*For the attention layer, when $L \leq L_0$ we have*

$$\langle E_1^w - E_2^w, W_3 E_L^w \rangle \geq \Omega\left(\frac{\eta^2}{L}\right), \quad \forall w,$$

$$\langle E_1^{-w} - E_2^w, W_3 E_L^w \rangle \leq -\Omega\left(\frac{\eta^2}{L}\right), \quad \forall w,$$

$$\langle E_2^w - E_\ell^w, W_3 E_L^w \rangle \geq \Omega\left(\frac{\eta^2}{L}\right), \quad \forall \ell \geq 3, \forall w.$$

*When $L > L_0$, let $\ell_0 = L - L_0 + 1$ we have*

$$\langle E_{\ell_0}^{-w} - E_\ell^w, W_3 E_L^w \rangle \geq \Omega\left(\frac{\eta^2}{L}\right), \quad \forall \ell \neq \ell_0, \forall w,$$

$$\langle E_\ell^w - E_{\ell_0}^w, W_3 E_L^w \rangle \geq \Omega\left(\frac{\eta^2}{L}\right), \quad \forall \ell \neq \ell_0, \forall w,$$

$$\langle E_1^w - E_\ell^w, W_3 E_L^w \rangle \geq \Omega\left(\frac{\eta^2}{L}\right), \quad \forall \ell \neq \ell_0, 1, \forall w.$$

*Proof.* Since $J_{(n,0)} = \frac{1}{2}$, and $\varphi_\ell^{(n,0)} = \frac{1}{L}$ for $n \in I_L$ (the length is $L$), we have

$$\langle u_1, E_1^w \rangle = \eta \langle -\nabla_u \mathcal{L}_0, E_1^w \rangle$$
$$= \eta \sum_{L=1}^{L_{\max}} \frac{1}{2^L} \sum_{n \in I_L, \mathtt{C}(x_\ell^{(n)})=w} (-J'_{(n,0)}) y_n \varphi_1^{(n,0)}$$
$$= \frac{\eta}{4},$$

where the last equality is due to the cancellation between positive and negative samples whose length is greater than 1. Due to the same reason, for any $\ell \geq 2$, we have

$$\langle u_1, E_\ell^w \rangle = \eta \langle -\nabla_u \mathcal{L}_0, E_\ell^w \rangle$$
$$= \eta \sum_{L=2}^{L_{\max}} \frac{1}{2^L} \sum_{n \in I_L, \mathtt{C}(x_\ell^{(n)})=w} (-J'_{(n,0)}) y_n \varphi_1^{(n,0)}$$
$$= 0$$

Regarding the attention, for any token $E_\ell^{w'}$, we have

$$\langle E_\ell^{w'}, W_1 E_L^w \rangle = \eta \lambda \langle E_\ell^2, (-\nabla_W \mathcal{L}_1) E_L^w \rangle$$
$$= \frac{1}{2^L} \sum_{n \in I_L, \mathtt{C}(x_\ell^{(n)})w', \mathtt{C}(x_L^{(n)})=w} J'_{(n,0)} y_n \varphi_\ell^{(n,0)} \left( u_0^\top x_\ell^{(n)} - \mathtt{T}_0(X^{(n)}) \right)$$

$$= 0,$$

where the last inequality is due to the fact that $u_0 = 0$.

In summary, similar to what happens in even pairs problem, at time step 1, only the token score at the first position increases, and all other token scores remain 0, and the attention scores are all 0, resulting $\varphi_\ell^{(n,1)} = \frac{1}{L}$ for $n \in I_L$. Note that, we also have

$$-J'_{(n,1)} = \frac{1}{1 + \exp(\frac{\eta}{4L})}, n \in I_L^+; \quad -J'_{(n,1)} = \frac{1}{1 + \exp(-\frac{\eta}{4L})}, n \in I_L^-.$$

Next, we characterize the token scores and attention scores at time step 2.

$$\langle u_2, E_1^w \rangle = \langle u_1, E_1^w \rangle + \eta \langle -\nabla_u \mathcal{L}_1, E_1^w \rangle$$

$$= \frac{\eta}{4} + \eta \sum_{L=2}^{L_{\max}} \frac{1}{2^L} \sum_{n \in I_L, \mathsf{C}(x_\ell^{(n)})=w} (-J'_{(n,0)}) y_n \varphi_1^{(n,0)}$$

$$= \frac{\eta}{4} + \frac{\eta}{2} \frac{1}{1 + \exp(\eta/4)} + \sum_{L=2}^{L_{\max}} \frac{\eta}{2} \left( \frac{1}{1 + \exp(\frac{\eta}{4L})} - \frac{1}{1 + \exp(-\frac{\eta}{4L})} \right) \frac{1}{L}$$

Thus,

$$\left| \langle u_2, E_1^w \rangle - \frac{2\eta}{4} \right| \leq \frac{\eta}{2} \left| \frac{1}{1 + \exp(\eta/4)} - \frac{1}{2} \right| + \sum_{L=2}^{L_{\max}} \frac{\eta}{2L} \cdot \frac{\eta}{2L}$$

$$\leq \frac{3\eta^2}{8},$$

where the last inequality is due to the Lipchitz continuity of $f(x) = 1/(1+e^x)$ (with Lipchitz constant 1) and $\sum_{L=2}^{\infty} 1/L^2 \leq 1$.

Similarly, for $\ell \geq 2$, we have

$$\langle u_2, E_\ell^w \rangle = \eta \langle -\nabla_u \mathcal{L}_1, E_\ell^w \rangle$$

$$= \eta \sum_{L=\ell}^{L_{\max}} \frac{1}{2^L} \sum_{n \in I_L, \mathsf{C}(x_\ell^{(n)})=w} (-J'_{(n,1)}) y_n \varphi_\ell^{(n,0)}$$

$$= \sum_{L=\ell}^{L_{\max}} \frac{\eta}{2} \left( \frac{1}{1 + \exp(\frac{\eta}{4L})} - \frac{1}{1 + \exp(-\frac{\eta}{4L})} \right) \frac{1}{L}$$

$$\leq - \sum_{L=\ell}^{L_{\max}} \frac{\eta^2}{4L^2}$$

$$\leq - \frac{\eta^2}{4\ell^2}.$$

In addition, regarding the difference of token scores, we have

$$\langle u_2, E_2^w - E_\ell^2 \rangle = \sum_{L=2}^{L_{\max}} \frac{\eta}{2} \left( \frac{1}{1 + \exp(\frac{\eta}{4L})} - \frac{1}{1 + \exp(-\frac{\eta}{4L})} \right) \frac{1}{L} - \sum_{L=\ell}^{L_{\max}} \frac{\eta}{2} \left( \frac{1}{1 + \exp(\frac{\eta}{4L})} - \frac{1}{1 + \exp(-\frac{\eta}{4L})} \right) \frac{1}{L}$$

$$= \sum_{L=2}^{\ell-1} \frac{\eta}{2} \left( \frac{1}{1 + \exp(\frac{\eta}{4L})} - \frac{1}{1 + \exp(-\frac{\eta}{4L})} \right) \frac{1}{L}$$

$$\leq -\frac{\eta^2}{16},$$

where $\ell \geq 3$.

Therefore, we already prove that

$$\langle u_2, E_1^w \rangle = \Theta(\eta t), \quad \forall w,$$
$$\langle u_2, E_\ell^w \rangle = -\Theta(\eta^2 t^2), \quad \forall \ell \geq 2, \forall w,$$
$$\langle u_2, E_2^w - E_\ell^w \rangle \leq -\Omega(\eta^2 t^2), \quad \forall \ell \geq 3, \forall w.$$

Next, we analyze the attention score. For any length $L < L_0$, the proof follows the same steps in Lemma B.2. Here, we just present the results.

$$\lambda \langle E_1^w - E_2^w, (-\nabla_W \mathcal{L}_1) E_L^w \rangle \geq \Omega \left( \frac{\eta}{L} \right),$$
$$\lambda \langle E_1^{-w} - E_2^w, (-\nabla_W \mathcal{L}_1) E_L^w \rangle \leq -\Omega \left( \frac{\eta}{L} \right).$$

In addition,

$$\lambda \langle E_1^w - E_2^w, (-\nabla_W \mathcal{L}_2) E_L^w \rangle \geq \Omega \left( \frac{\eta}{L} \right)$$
$$\lambda \langle E_1^{-w} - E_2^w, (-\nabla_W \mathcal{L}_2) E_L^w \rangle \leq -\Omega \left( \frac{\eta}{L} \right),$$

which implies that for any $L \leq L_0$, we have

$$\langle E_1^w - E_2^w, W_3 E_L^w \rangle \geq \Omega \left( \frac{\eta^2}{L} \right)$$
$$\langle E_1^{-w} - E_2^w, W_3 E_L^w \rangle \leq -\Omega \left( \frac{\eta^2}{L} \right),$$

**Next, we show that the initial dynamics of attention scores in sequence with length $L > L_0$.**

Recall that $\ell_0 = L - L_0 + 1$. We have

$$\lambda \langle E_{\ell_0}^{-w} - E_1^w, (-\nabla_W \mathcal{L}_1) E_L^w \rangle = -\frac{1}{2^L} \sum_{n \in I_L^-} (-J'_{(n,1)}) \varphi_{\ell_0}^{(n,1)} \left( \langle u_1, E_{\ell_0}^w \rangle - \frac{\eta}{4L} \right)$$
$$- \frac{1}{2^L} \sum_{n \in I_L, \mathsf{C}(x_1^{(n)}) = w} (-J'_{(n,1)}) y_n \varphi_1^{(n,1)} \left( \langle u_t, E_1^w \rangle - \frac{\eta}{4L} \right)$$
$$= \frac{1}{8} \cdot \frac{1}{1 + \exp(-\frac{\eta}{4L})} \cdot \frac{1}{L} \left[ -2 \langle u_1, E_{\ell_0}^w \rangle + \langle u_1, E_1^w \rangle + \frac{\eta}{4L} \right]$$
$$- \frac{1}{8} \frac{1}{1 + \exp(\frac{\eta}{4L})} \cdot \frac{1}{L} \left( \langle u_1, E_1^w \rangle - \frac{\eta}{4L} \right)$$
$$\overset{(a)}{\geq} \Omega \left( \frac{\eta}{L} \right),$$

where $(a)$ is due to the Lipchitz continuity of $f(x) = 1/(1 + e^x)$ and $\langle u_1, E_1^w \rangle > \Omega(\eta)$.

Similarly,

$$\lambda \langle E_{\ell_0}^w - E_1^w, (-\nabla_W \mathcal{L}_1) E_L^w \rangle \leq -\Omega \left( \frac{\eta}{L} \right),$$

Since $\langle u_1, E_\ell^w \rangle = 0$ for all $\ell \geq 2$, we have

$$\lambda \langle E_\ell^w - E_{\ell'}^w, (-\nabla_W \mathcal{L}_1) E_L^w \rangle = 0.$$

Finally, we aim to show that at time step $t = 3$, the attention layer also distinguishes between first and other tokens. This can be done by noting that $\langle u_2, E_1^w - E_\ell^2 \rangle \geq \Omega(\eta)$ for $\ell \neq \ell_0$. More importantly, $\varphi_1^{(+,2)} > \varphi_1^{(-,2)}$ Specifically, we have

$$
\begin{aligned}
&\eta\lambda \langle E_1^w - E_\ell^w, (-\nabla_W \mathcal{L}_2) E_L^w \rangle \\
&= \frac{\eta}{2^L} \sum_{n \in I_L, \mathtt{C}(x_2^{(n)}) = \mathtt{C}(x_L^{(n)}) = w} (-J'_{(n,2)}) y_n \varphi_1^{(n,2)} \left( \langle u_2, E_1^w \rangle - \mathtt{T}_2(X^{(n)}) \right) \\
&\quad - \frac{\eta}{2^L} \sum_{n \in I_L, \mathtt{C}(x_\ell^{(n)}) = \mathtt{C}(x_L^{(n)}) = w} (-J'_{(n,2)}) y_n \varphi_\ell^{(n,2)} \left( \langle u_2, E_\ell^w \rangle - \mathtt{T}_2(X^{(n)}) \right) \\
&\stackrel{(a)}{=} \frac{\eta}{8} (-J'_{(+,t)}) \left[ \varphi_1^{(+,2)} \langle u_2, E_1^w \rangle - \varphi_\ell^{(+,2)} \langle u_2, E_\ell^w \rangle \right] \\
&\quad - \frac{\eta}{8} (-J'_{(-,t)}) \left[ \varphi_1^{(+,2)} \langle u_2, E_1^w \rangle - \varphi_\ell^{(+,2)} \langle u_2, E_\ell^w \rangle \right] \\
&\geq \Omega(\eta^4/L),
\end{aligned}
$$

where $(a)$ is due to the fact that $\varphi_\ell^{(n,2)}$ are equal for any $\ell \geq 2$ and $\ell \neq \ell_0$, and the last inequality follows from that $\langle u_2, E_1^w \rangle - \langle u_2, E_\ell^w \rangle \geq \Omega(\eta)$.

Thus, the proof is complete. $\qquad\square$

For the rest of the proof, the steps follow the same as in Appendix B. Specifically, by induction, we have

**Theorem C.3** (Phase 1). *[Restatement of Theorem 5.1] Let $-w$ denote the flip of token $w \in \{\mathtt{a}, \mathtt{b}\}$. Choose $\lambda = \Omega(L_{\max}^2)$ and $t_0 = O(1/(\eta L_{\max}))$. Then, for all $t \leq t_0$, the parameters evolve as follows:*

*(1) The dynamics of linear layer $u$ is governed by the following inequalities.*

$$
\begin{aligned}
\langle u_t, E_1^w \rangle &= \Theta(\eta t), \quad \forall w, \\
\langle u_t, E_\ell^w \rangle &= -\Theta(\eta^2 t^2), \quad \forall \ell \geq 2, \forall w, \\
\langle u_t, E_2^w - E_\ell^w \rangle &\leq -\Omega(\eta^2 t^2), \quad \forall \ell \geq 3, \forall w.
\end{aligned}
$$

*(2) The dynamics of the attention layer $W$ is governed by the following inequalities. For any length $L < L_0$, we have*

$$
\begin{aligned}
\langle E_1^w - E_\ell^{w'}, W_t E_L^w \rangle &\geq \Omega(\eta^2 t^2), \quad \forall \ell \geq 2, \forall w, w', \\
\langle E_\ell^{w'} - E_1^{-w}, W_t E_L^w \rangle &\geq \Omega(\eta^2 t^2), \quad \forall \ell \geq 2, \forall w, w', \\
\langle E_2^{w'} - E_\ell^{w''}, W_t E_L^w \rangle &\geq \Omega(\eta^4 t), \quad \forall \ell \geq 3, \forall w, w', w''.
\end{aligned}
$$

*For length $L \geq L_0$, let $\ell_0 = L - L_0 + 1$, and we have*

$$
\begin{aligned}
\langle E_{\ell_0}^{-w} - E_\ell^{w'}, W_t E_L^w \rangle &\geq \Omega(\eta^2 t^2), \quad \forall \ell \neq \ell_0, \forall w, w', \\
\langle E_\ell^{w'} - E_{\ell_0}^{-w}, W_t E_L^w \rangle &\geq \Omega(\eta^2 t^2), \quad \forall \ell \neq \ell_0, \forall w, w', \\
\langle E_1^{w'} - E_\ell^{w''}, W_t E_L^w \rangle &\geq \Omega(\eta^4 t), \quad \forall \ell \neq 1, \ell_0, \forall w, w', w''.
\end{aligned}
$$

Therefore, at the end of phase 1, i.e., $t \leq t_0$, the dataset $\{(\sum_{\ell=1}^{L} x_\ell^{(n)} \varphi_\ell^{(n,t_0)}, y_n)\}$ is separable. This fact enable us to performa the same implicit bias analysis as in Appendix B. Thus, we conclude that similar theorems Theorem 5.2 and Theorem 5.3 hold true.

# D. Additional Experiments

**Imapct of the scaling parameter $\lambda$.** We first select two additional $\lambda$ configurations for training on the Even Pairs task to demonstrate that the training dynamics remain consistent with those reported in the main paper.

**Two-phase dynamics.** As shown in Figure 6, the two-phase phenomenon naturally emerges even when a fixed step size is used throughout the entire gradient descent training process. Notably, this behavior also appears in real-world datasets: Figure 7 demonstrates that the two-phase learning dynamics persist when training with NanoGPT (Karpathy, 2023).

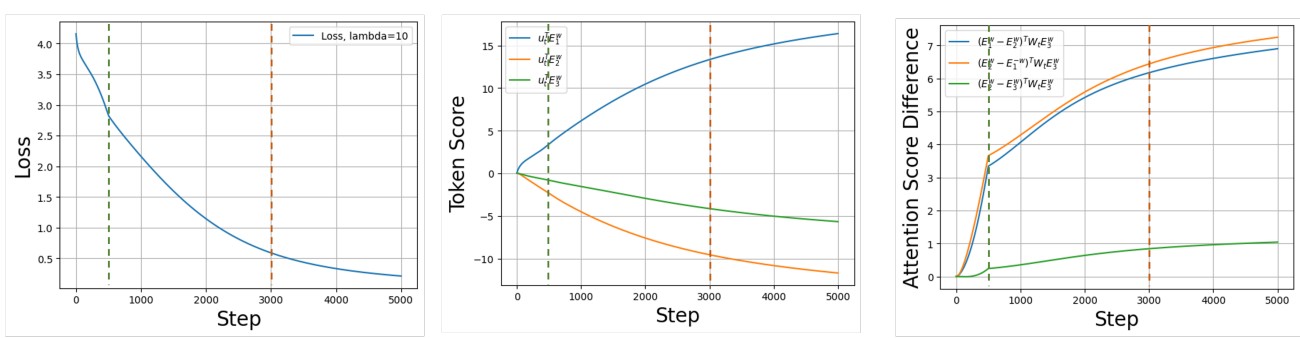

*Figure 4.* Training dynamics of learning Even pairs when $\lambda = 10$

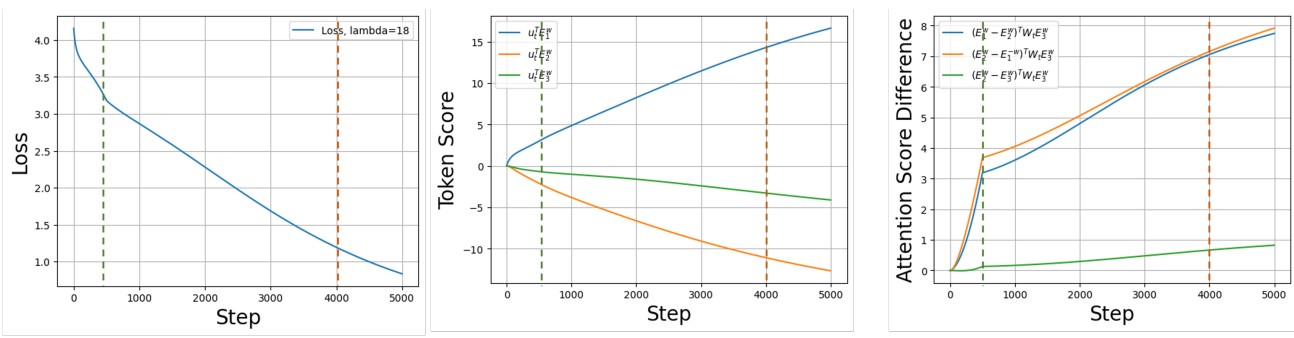

*Figure 5.* Training dynamics of learning Even pairs when $\lambda = 18 = L_{\max}^2/2$

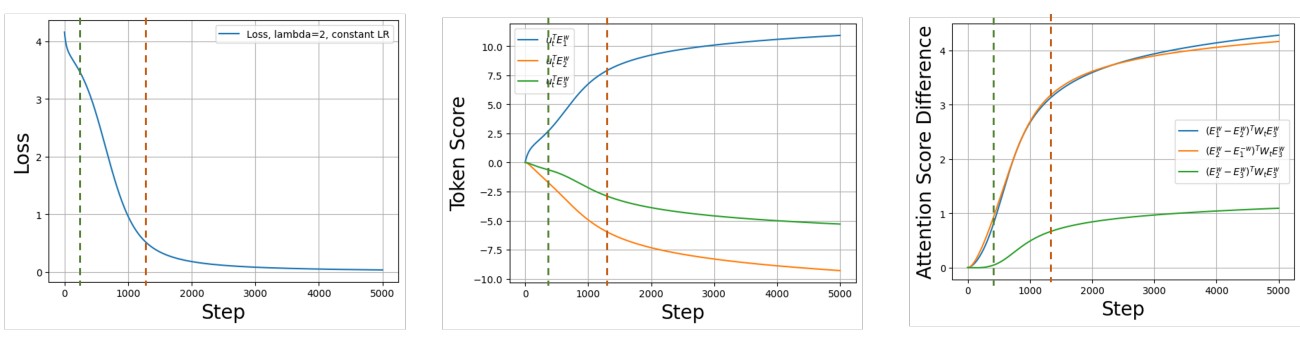

*Figure 6.* Training dynamics of training transformer on even pairs using vanilla GD (constant learning rate) and $\lambda = 2$.

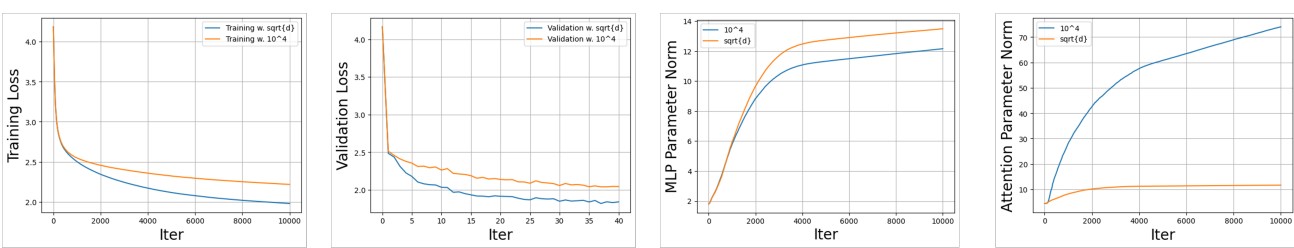

*Figure 7.* Results of nanoGPT, trained on 'shakespeare' dataset. Configuration: block-size=64, batch-size=12, n-layer=4, n-head=4, n-embd=128

