# OpenReview forum: "How Transformers Learn Regular Language Recognition: A Theoretical Study on Training Dynamics and Implicit Bias"
_ICML.cc/2025/Conference — ICML 2025 poster_

### Official Review · Reviewer_fgjp · 2025-03-10

**Overall Recommendation:** 2

**Summary:**

The work shows a theoretical analysis studying how a single layer of a Transformer (more precisely, an attention layer with a linear layer on top) learns to solve "even pairs" and "parity check" - two regular language recognition tasks. The authors begin by analyzing the even pairs task, showing that the Transformer trained in two phases learns to solve the even pairs problem, and discuss the training dynamics. Next, the authors discuss how parity check can be solved with Chain-of-Thought (CoT), either by inference with a trained Transformer or by training with CoT data. The authors verify their results experimentally.

**Claims And Evidence:**

I think that the theoretical analysis of Transformers trained on the tasks studied in the paper is novel, and the results are interesting in the context of understanding training dynamics and capabilities of Transformers for solving language learning tasks. However, there are some flaws in the paper that I believe the authors need to address:
- My main concern is that, beyond showing nice theoretical analysis and some study of training dynamics, it is unclear what is the main takeaway from the paper. What do these results teach us about what Transformers can do, or about how they learn, that we didn't know before? One conclusion that is addressed in the discussion is the importance of CoT for learning complex tasks like parity check. However, this seems to overlap with results shown in previous works, and it is unclear to me how the analysis of parity learning with CoT shown in this work differs from previous results of similar flavor. Another interesting result is that learning "even pairs" can help with learning parity check, but this is not shown directly and not discussed. I believe that stating clearly what is the main novel conclusion from the theoretical analysis and what it informs us about learning with Transformers can greatly improve the paper.
- Related to the above, I believe that the relation between learning "even pairs" and "parity check" is interesting, but it is not studied in a setting that captures learning with language models. Specifically, the result for using transformers trained on even pairs to compute parity during inference seems particularly synthetic, and it is unclear whether this is just a way to introduce the next result in the section, or whether this results is interesting on its own. For the analysis of the second approach, adding the regularization with respect to the even pairs appears synthetic and it is unclear why it is needed. Can a similar result be shown by just changing the data mixture (e.g., training on a mixture of "even pairs" and "parity check" task, and testing on "parity check" with CoT)? How does the "even pairs" regularization improves training, given other results showing that parity learning is possible just with CoT?
- The authors emphasize that learning happens in two phases, but it seems that the phases arise from the change of hyper-parameters at some point in training. While this is reasonable for the theoretical analysis, I believe that the authors should discuss whether the two-phase learning should happen with a standard learning-rate schedule. In particular, I think that experimental results in a standard setting (standard, maybe even deep, Transformer, standard learning rate schedule etc.) would be helpful.
- Additionally, I think that connecting the theoretical conclusions to real-world problems (even somewhat synthetic problems) through additional experiments could be a good way to make this work more appealing to a broader audience.

To summarize, I believe that the theoretical setting and results are interesting, but the bottom-line conclusion is unclear, and some of the results are perhaps too synthetic.

**Essential References Not Discussed:**

No.

**Experimental Designs Or Analyses:**

See above.

**Methods And Evaluation Criteria:**

See above.

**Other Comments Or Suggestions:**

N/A

**Other Strengths And Weaknesses:**

N/A

**Questions For Authors:**

See above.

**Relation To Broader Scientific Literature:**

See above.

**Theoretical Claims:**

See above.

---

> ### Author Rebuttal · Authors · 2025-04-01
>
> We thank the reviewer for the helpful comments. Please note that all our new experiment results (i.e., the figures we quote below) can be accessed via the link https://anonymous.4open.science/r/icml2025-4BCC/Figures%20for%20ICML.pdf
>
> Q1: Main takeaway about transformers: (i) How analysis of parity learning with CoT differs from previous results; (ii) How learning "even pairs" help with learning of parity check.
>
> A1: **(i)** There are two key differences of our analysis from previous theoretical analysis [1,2] of parity check. *First*, [1,2] analyze only the first one or three gradient descent steps, not the entire training process. They do not prove the convergence of the training algorithm as the number of iterations goes large. In contrast, we characterize the dynamics of the entire training process, with the loss converging to global minimum (nearly zero) (Theorems 5.1–5.3), which is a comprehensive analysis of training process. *Second*, prior studies requires all input data sequence to have the same length, whereas our approach allows inputs to have  different lengths -- a more realistic setting. Such a more general setting is much more challenging to analyze, because tokens appearing in different sequence lengths influence each other’s gradient updates. Our analysis explicitly captures these dependencies and provides a fine-grained analysis of individual token values, making our technique more generalizable to broader learning scenarios.
>
> **(ii)** To connect the two problems, the learning output of "even pairs" serves as the first step in CoT towards solving parity check. Hence, using "even pairs" as a regularizer in parity check loss can provide an initial momentum to start the training of parity check. The reviewer can refer to lines 330–347 right column to see more detailed explanation in terms of how parameter updates.
>
> **Further insights:** (a) Layer roles: Our analysis characterizes distinct roles of feed-forward and attention layers via their joint training dynamics. Specifically, the attention layer $W$ learns to capture token-level semantics—e.g. token equality—while the linear layer $u$ encodes positional information. These functional separations are expected to persist in deeper architectures, where different layers may specialize to recognize different information such as content or positional information. (b) Training process exhibits different phases, where the first phase is parameter alignment and the second phase enables growing parameter norm and loss fast decay. Such dynamics is also observed in real-world problems (see Figure 3). (c) Our developed techniques for analyzing joint training of linear and attention layers can be useful for studying more general transformers.
>
> [1] Kim, J. and Suzuki, T. Transformers provably solve parity efficiently with chain of thought. ICLR 2025
>
> [2] Wen et al. From Sparse Dependence to Sparse Attention: Unveiling How Chain-of-Thought Enhances Transformer Sample Efficiency, arXiv 2024
>
> Q2: (i) It is unclear whether Algorithm 1 is interesting on its own. (ii) Why adding the regularization is needed, and how does the "even pairs" regularization improves training.
>
>
> A2: **(i):** Our Approach 1 in Section 5.1 is new and of independent interest. Existing works studying CoT typically require training with CoT data [1,2]. In contrast, our Approach 1 shows that transformers trained on even pairs can solve parity check by simply using CoT at **inference time** without additional training.
>
> **(ii):** The output of "even pairs” algorithm serves as the first step in CoT towards solving parity check. Hence, using "even pairs" as regularization provides initial momentum to the training process for parity check. Also note that such a regularization approach is equivalent to changing the data distribution, i.e., mixing the even pairs data and parity data together and training the transformer model.
>
> Q3: Does the two-phase learning happen with a standard learning-rate schedule?
>
> A3: Yes. For vanilla GD with fixed stepsize throughout the training process, our new experiments (Figure 4) show that the training still exhibits the similar two-phase process as characterized in our theorem.
>
> Q4: Connecting theoretical conclusions to real-world problems through additional experiments
>
> A4: We provide new experiments in Figure 3 on real-world dataset 'shakespeare' with deeper and realistic transformers (nanoGPT). It can be observed that the training  with vanilla Adam optimizer also exhibits a two-phase learning curve, i.e., the parameter norm grows fast in phase 1 and slows down in phase 2.
>
> **Generalization:** Our analysis framework can be extended to study general regular language problems. More details are provided in A4 of our response to Reviewer TZ3J.
>
> Thank you again for your insightful comments. We hope our responses addressed your concerns and would greatly appreciate your kind consideration in increasing your score.

---

### Official Review · Reviewer_yZje · 2025-03-12

**Overall Recommendation:** 3

**Summary:**

This paper presents a detailed theoretical analysis of how a one-layer transformer learns two sequence recognition/classification tasks: even pairs and parity check. The analysis decomposes the factors driving the attention weights and token score, with a discussion of the training dynamics in detail (e.g. attention weight reliance on different tokens, linear layer performing a max-margin solution).

**Claims And Evidence:**

They seem fine overall.

**Essential References Not Discussed:**

Not to my knowledge

**Experimental Designs Or Analyses:**

The experiment in Section 6 makes sense.

**Methods And Evaluation Criteria:**

The two tasks seem appropriate and standard from the mentioned related work.

**Other Comments Or Suggestions:**

Related to the accessibility point above, I think it would be great if the authors could add more diagrams, or interleave the theoretical analyses with the empirical results. These can help a more general reader understand the work and in turn interest a broader audience.

**Other Strengths And Weaknesses:**

This work is a bit outside of my expertise so I'll comment from a general perspective. The biggest concern I have is the accessibility of this work to a broader audience. While I get the gist of ideas, I find the paper a bit hard to follow.

**Questions For Authors:**

It was unclear to me if the two-phase phenomena is entirely emergent or tied to the two-phase gradient descent schedule, which from Figure 3 I take to be the former (emergent).

**Relation To Broader Scientific Literature:**

This work extends prior theoretical work on the learning dynamics of transformers on a more representative architecture and on multiple tasks.

**Theoretical Claims:**

I found sections 4 and 5 a bit hard to follow, see notes below.

---

> ### Author Rebuttal · Authors · 2025-04-01
>
> We thank the reviewer for the helpful comments. Please note that all our new experiment results (i.e., the figures we quote below) can be accessed via the link  https://anonymous.4open.science/r/icml2025-4BCC/Figures%20for%20ICML.pdf
>
> Q: Add more diagrams, or interleave the theoretical analyses with the empirical results.
>
> A: Thanks for the suggestion. We will add a diagram to illustrate what each training phase of the even pairs problem learn, and will add a diagram to illustrate how the CoT-based inference approach solves the parity check problem. We will also present the experiments of the learning phases for even pairs and parity check problems right after theorems for their training dynamics.
>
> Q: It was unclear to me if the two-phase phenomena is entirely emergent or tied to the two-phase gradient descent schedule, which from Figure 3 I take to be the former (emergent).
>
> A: Yes. As demonstrated by our additional experiments (Figure 4), the two-phase phenomena arises naturally even under a fixed stepsize for the entire gradient descent training process.
>
> Thank you again for your insightful comments. We hope our responses addressed your concerns and would greatly appreciate your kind consideration in increasing your score.

---

> > ### Comment · Reviewer_yZje · 2025-04-08
> >
> > Thank you for your response. I would like to keep my score and recommend the area chair to consider the feedback provided by reviewers with more direct expertise when making the final decision.

---

> > > ### Author Response · Authors · 2025-04-09
> > >
> > > Thank you very much for taking time to read our rebuttal and for your thoughtful comment. We sincerely appreciate your positive opinion!

---

### Official Review · Reviewer_btPg · 2025-03-12

**Overall Recommendation:** 4

**Summary:**

The authors theoretically study language recognition tasks with transformer. Formally, they study the training dynamics of transformers trained with gradient descent on the parity check and even pairs problems. Considering a single-layer simplified transformer, they first show that the even pairs problem can be solved directly before showing how Chain-of-Thought was needed for the harder parity check problems, either at inference with a model pre-trained on even pairs or during the training itself. For both tasks, the authors identify two distinct phases of learning where the attention grows in the first, mapping data to separable vectors. Then, once the attention module is stable, the linear layer grows and implements ma ax-margin decision boundary to separate the attention outputs between positive and negative samples. They provide a convergence rate for the training loss and empirically validate their findings on synthetic data.

**Claims And Evidence:**

The theoretical claims are well supported by clear and detailed proofs, and the authors also provide experimental validation of their theory in a controlled setting.

**Essential References Not Discussed:**

To the best of my knowledge, there are no essential references not discussed in the current submission.

**Experimental Designs Or Analyses:**

The experimental setup is well explained and remains in a controlled setting. I believe the current submission would be strengthened with additional visualization to illustrate the different training dynamics and phases identified by the authors.

**Methods And Evaluation Criteria:**

The main contributions of this work are theoretical, and the authors provided detailed proofs and brief experimental computation to validate the theory in a synthetic setting. I believe the methods and evaluation criteria make sense for the problem.

**Other Comments Or Suggestions:**

I list below some potential typos:

- l. 121, second column: "Frobenious" --> "Frobenius"
- l 400, first column: "thee" --> "the"

**Other Strengths And Weaknesses:**

**Strengths**

- The paper is very well-written and motivated
- The problem tackled is of great interest
- I appreciate that the model considered is (slightly) more general than prior works [1], which leads to a more involved analysis
- The findings are interesting and well explained to grasp their impact on our understanding of the training dynamics
- The proofs of clear and elegant

**Weakness**

I list below what I think are weaknesses, but I would be happy to be corrected if I misunderstood some important aspects of the authors' contributions.

- While the authors' approach is indeed more general thatn [1] because of the linear layer, it is still less general that [2] since the linear layer is incorporated the value matrix $W_V$ in the optimization which amounts to simply consider an attention module and not a transformer: typically feed-forward blocks have non-linearities as it was considered in [2].
- I find some part of the setting oversimplified compared to more practical settings and some of these simplifications are not well-motivated in my opinion (see questions).
- The current submission lacks additional visualization or empirical validation since the considered setting seems a little bit ad-hoc and remains simple. It would be interesting to see how the figures vary with other hyperparameters for the controlled setting.

Overall, I find the paper interesting and the analysis well conducted. I believe this is valuable work that can be improved with more visualization and variety in the controlled setting to illustrate the theoretical findings better. This is the reason for my current score, but I remain open to modifying my score, provided the authors clarify the points mentioned in the weaknesses section.

*References*

[1] Kim, J. and Suzuki, T. Transformers provably solve parity efficiently with chain of thought. ICLR 2025

[2] Wen et al. From Sparse Dependence to Sparse Attention: Unveiling How Chain-of-Thought Enhances Transformer Sample Efficiency, arxiv 2024

**Update after rebutal**: increased score from 3 to 4.

**Questions For Authors:**

1) It seems that the embedding strategy described from l.171 only depends on the position in the sequence and not on the token value. Could the author clarify this point? If this is the case, then I believe it is not very realistic since for the tasks considered in the submission, the value of the token matters a lot (and it is, of course, the case in more practical settings).

2) In Theorem 41, the authors propose to choose $\lambda = \Omega(L_{max}^2)$ instead of the common $\sqrt{d}$ in the vanilla transformer (motivated to avoid small gradients when the magnitude of logits increases). However, the order of $L_{max}$ and $d$ in common transformers is typically $\approx 10^2$ (time series forecasting [1], ViT [2], BERT [3], etc.). It means replacing a scaling of order $10^1$ by a scaling of order $10^4$. I believe this will lead experimentally to shrinking the logits range and outputting almost uniform attention values. Could the authors elaborate on that?

3) Related to the point above, I notice that in their experiments, the authors take $L_{max} = 6$, which seems quite small, given again the type of sequences processed by transformers in more practical settings, and $\lambda=2$. This does not seem to match the predicted $\lambda=\Omega(L_{max}^2)$ of Theorem 4.1. Maybe I misunderstood the setting, but could the authors clarify that point?

4) I am intrigued by the gradient descent in two phases considered in the analysis. Why did the authors choose this framework instead of a more classical GD, SGD, or more adapted Adam (noting that adaptive optimizers are better for transformer and attention-based models [4])?

5) Related to the point above, how did the author select the value $t_0$? It seems that the optimization can be heavily impacted because of that, especially given that the parameter $\lambda$ is used in front of the learning rate. Could the authors elaborate on that?

*References*

[1] Ilbert et al. SAMformer: Unlocking the Potential of Transformers in Time Series Forecasting with Sharpness-Aware Minimization and Channel-Wise Attention, ICML 2024

[1] Dosovitskiy et al., An Image is Worth 16x16 Words: Transformers for Image Recognition at Scale, ICLR 2021

[3] Devlin et al. BERT: Pre-training of Deep Bidirectional Transformers for Language Understanding, NAACL 2019

[4] Zhang et al. Why are adaptive methods good for attention models? NeurIPS 2020

**Relation To Broader Scientific Literature:**

I find that related work and prior works are well introduced and compared. The submission's contributions are interesting and are part of a growing interest in the literature on the theoretical understanding of transformers. The novelty seems to come mostly from the slightly more general model considered and the study of language recognition tasks with truncated CoT framework. The max-margin solution seems connected to [1] that showed how gradient-descent on softmax model converges in direction to a max-margin solution that separates
locally-optimal tokens from non-optimal ones.

[1] Tarzanagh et al. Max-Margin Token Selection in Attention Mechanism, NeurIPS 2023

**Theoretical Claims:**

The theoretical findings are very detailed and greatly explain the training dynamics. The proofs are well thought, detailed, and clear. In my opinion, these are the main strengths of this work.

---

> ### Author Rebuttal · Authors · 2025-04-01
>
> We thank the reviewer for the helpful comments. Please note that all our new experiment results (i.e., the figures we quote below) can be accessed via the link https://anonymous.4open.science/r/icml2025-4BCC/Figures%20for%20ICML.pdf
>
> Q1: It seems that the embedding strategy does not depend on the token value.
>
> A1: We would like to clarify that the embedding strategy does depend on both the token value and the position. To see this, suppose the token is at the $\ell$-th position in the sequence. Then if the token is symbol 'a', embedding vector $E$ has 1 at the $\ell$-th **odd** index, i.e., $E[2\ell-1]=1$; otherwise if the token value is 'b', the embedding vector has 1 at the $\ell$-th **even** index, i.e., $E[2\ell]=1$. Clearly, embedding also depends on the token value.
>
> Q2: In experiments, authors take $L_{\max}=6$ and $\lambda=2$, which does not match $\lambda=\Omega(L_{\max}^2)$ in Theorem 4.1.
>
> A2: We provide new experimental results (see Figures 1 and 2) under $\lambda=L_{\max}^2$, which exhibit similar training phases as our original experiments and validate our theory. We further note that $\lambda=\Omega(L_{\max}^2)$ of Theorem 4.1 is a sufficient condition to guarantee the convergence of the training theoretically, but may not be necessary. In practice, as  demonstrated in our original experiment, a much smaller $\lambda=2$ can also lead to desired convergence even with a faster rate.
>
> Q3: Theorem 4.1 chooses $\lambda=\Omega(L_{\max}^2)$ instead of the common $\sqrt{d}$ in the vanilla transformer. Will this lead experimentally to shrinking the logits range and outputting almost uniform attention values?
>
> A3: As discussed in A2, $\lambda = \Omega(L_{\max}^2)$ achieves the same desirable output (non-uniform attention) as smaller $\lambda$, although the training becomes slower. We also provide new experiments on deeper and larger models (see Figure 3), and observe that choosing a large scaling factor may not always degrade the performance drastically. Several reasons can possibly explain this. (a) Deeper models have layer normalization and adaptive learning rates, which might compensate for smaller gradients. (b) The dynamics of feed-forward layers can help activate softmax to learn so that attention is not uniform. (c) The attention weights will learn to increase its value to mitigate the high scaling factor (as shown in the Figure 3). We will include those experiments and discussion on the choice of $\lambda$ in the paper.
>
> Q4: I am intrigued by the gradient descent in two phases considered in the analysis. Why did the authors choose this framework instead of a more classical GD, SGD, or more adapted Adam (noting that adaptive optimizers are better for transformer and attention-based models [4])?
>
> A4: Our two-phase learning-rate schedule serves as a simplified approximation of the decaying learning rate in Adam, which takes high learning rate in early training and then takes smaller learning rate in later training. See lines 182-191 (right column) for more detailed explanation.
>
> Nevertheless, even for classical GD with fixed stepsize throughout the training process, our new experiments (Figure 4) show that the training still exhibits the similar two-phase process as characterized in our theorem.
>
> Q5: How did the author select the value $t_0$?
>
> A5: In Theorem 4.1, we provide a parameter setting that guarantees a desirable training output, where $t_0 = 1/(\eta L_{\max})$, and $\eta = O(\min\{1/L_{\max}, 1/\lambda^{2/3}\})$. For example, if we choose $\eta=1/\lambda$ and $\lambda=L_{\max}^2$, making the learning rate at the first phase $O(1)$, then $t_0=O(L_{\max})$. This choice suggests how to set the duration of the warm-up phase in practice, which can be chosen based on the length of input sequences.
>
> Thank you again for your insightful comments. We hope our responses addressed your concerns and would greatly appreciate your kind consideration in increasing your score.

---

> > ### Comment · Reviewer_btPg · 2025-04-06
> >
> > I thank the authors for their detailed answer and for the additional experiments. I appreciate the authors' efforts to address my concerns. I will consider that along with the other reviews (and their responses) for my final recommendation.
> >
> > **Update** --> After carefully reading other reviews and the authors' answers to them, I decided to increase my score, given that most of my concerns are addressed. Although the setting is simplified, the analysis is well done, and additional experiments on larger models have been conducted, showcasing the same two-phase pattern.

---

> > > ### Author Response · Authors · 2025-04-09
> > >
> > > Thank you very much for taking time to read our rebuttal and for your thoughtful reconsideration of our work. We sincerely appreciate your positive feedback and strong support!

---

### Official Review · Reviewer_TZ3J · 2025-03-24

**Overall Recommendation:** 3

**Summary:**

This paper focuses on two typical regular languages: even pairs and parity check. The authors show that one-layer transformers can learn even pairs directly without CoT. For parity check, it is shown that one-layer transformers can learn it with CoT and with a small amount of data mixing with even pairs.

**Claims And Evidence:**

The theoretical claims are supported by clear theorems and proofs.

**Essential References Not Discussed:**

/

**Experimental Designs Or Analyses:**

The experiments on one-layer transformers are presented. The experimental designs are sound and valid.

**Methods And Evaluation Criteria:**

/

**Other Comments Or Suggestions:**

In the second approach to parity check, it would be better to phrase the method of adding even pairs as data mixing rather than adding a regularization loss, since data mixing is a more intuitive term to describe this process. I agree that its effect is similar to regularization, though.

**Other Strengths And Weaknesses:**

Strengths:
1. The theoretical setting is very neat and clean.
2. Many prior theoretical works train layer weights of transformers separately, which is not very natural. This paper directly analyzes the joint training of both layers.

Weaknesses:
1. It is not clear how the results, or even how this line of research, can be extended to deep transformers. In fact, Deletang et al. (2023) (cited in the paper) showed that deep transformers are not able to learn even pairs (acc is not 100%). It might be the case that the setting studied in this paper is just a delicate setting that happens to enable transformers to learn these two regular languages. If this is the case, it would be important to discuss how generalizable the results are and whether there is something to learn from this paper for deep transformers.
2. Although I really appreciate that the authors made the setting clean and neat, I do feel that the proof may critically depend on some seemingly weak assumptions: (1) zero initialization; (2) two-phase LR schedule; (3) merging $W_u W_v$ as $u$ and $W_kW_q$ as W.
3. The authors claim that one of their main technical contributions is to analyze the joint training of both layers. However, in the first phase of the two-phase LR schedule, the learning rate of the two layers is decoupled, which makes me suspect that the analysis may implicitly assume that one layer is trained much faster than the other one. In this case, the analysis is not really for the joint training of both layers.
4. I feel the analysis is too ad-hoc to the two regular languages studied in this paper. It is not clear how easily their analysis can be extended to other regular languages.

**Questions For Authors:**

1. I would like to see the authors' thoughts on Weakness 1. I would like to raise my score if the authors could point out some useful insight that can be extended to deep transformers.
2. I would like to see the authors' thoughts on Weaknesses 2, 3, 4. I would like to raise my score if the authors could actually show that the proof is flexible enough to hold with big variations of the setting.

**Relation To Broader Scientific Literature:**

It is crucial to understand whether transformers can learn formal languages defined according to Chomsky's hierarchy, and this has been studied in many prior works empirically or theoretically. This paper contributes to this line of research by proving that one-layer transformers can learn even pairs and parity check, the two most basic regular languages, under a neat theoretical setting.

**Theoretical Claims:**

Due to the large amount of theoretical papers I have been assigned to review, I cannot verify every proof in full detail. The theorem statements look correct to me.

---

> ### Author Rebuttal · Authors · 2025-04-01
>
> We thank the reviewer for the helpful comments. Please note that all our new experiment results (i.e., the figures we quote below) can be accessed via the link https://anonymous.4open.science/r/icml2025-4BCC/Figures%20for%20ICML.pdf
>
> Q1: It is not clear how the results can be extended to deep transformers
>
> A1: We thank the reviewer for the thoughtful comment. We acknowledge that there are several settings that make our results seem better than empirical study. E.g., we construct an orthonormal embedding vectors, whereas Deletang et al. (2023) needs to learn the embedding functions and may lead to additional errors. However, our results still offer several key insights for deep transformers as discussed below.
>
> **(i) Layer roles:** Our theoretical analysis uncovers distinct roles played by different components of the transformer during training. Specifically, the attention layer $W$ learns to capture token-level semantics—e.g. token equality—while the linear layer $u$ encodes positional information. These functional separations are expected to persist in deeper architectures, where different layers may specialize to recognize different information such as content or positional information. **(ii) From simple to complex tasks via CoT**: We theoretically justify how transformers trained on simple tasks can generalize to complex tasks using CoT (Algorithm 1). This helps explain the success of CoT in scaling model reasoning capabilities without additional training. **(iii) Stabilizing training with simple-task supervision:** We demonstrate that incorporating simple tasks data (e.g. even pairs) improve the initial training of complex tasks (e.g. parity check) using CoT. This finding provides practical guidance for constructing datasets in deep transformers—including simpler examples can stabilize training process.
>
> Q2: The proof may critically depend on some seemingly weak assumptions: (1) zero initialization; (2) two-phase LR schedule; (3) merging $W_u W_v$ as $u$ and $W_k W_q$ as $W$
>
> A2: We would like to point out that these assumptions can be relaxed or justified as follows.
>
> (1) can be relaxed to random initialization such as Gaussian initialization. Then concentration inequalities can ensure that initial parameters are relatively small with high probability. Then, similar techniques can be applied to analyze the training dynamics of transformers to show that GD updates will push parameters on the right track. (2) serves as a simplified approximation of the decaying learning rate in Adam, which takes high learning rate in early training and then takes smaller learning rate in later training. See lines 182-191 (right column) for more detailed explanation.
> (3) can be removed by separately training $W_k, W_q$, $W_v$ and $u$. Our analysis can be extended to include analyzing the changes of more terms such as $W_k W_k^\top$, $W_q^\top W_q$, and $\langle x_\ell W_k , W_q x_L\rangle$ during the training. We expect that the core idea remains similar.
>
> Q3: The analysis may not be really for the joint training of both layers.
>
> A3: Our analysis is indeed joint training. While the learning rates of the two layers are different in the first phase, our analysis does not de-couple their changes into two-time scales.
> Specifically, on pages 16-21, we explicitly analyze the effect of gradients of $u$ and $W$ on parameters update at *each time step* and take into consideration how one parameter affects another.
>
> Q4: I feel the analysis is too ad-hoc to the two regular languages studied in this paper.
>
> A4: Thanks for the insightful question. While our analysis focuses on two specific regular languages, the core techniques, token dependencies, and the role of CoT, are generalizable via the following unified view of regular language tasks.
>
> Every regular language can be recognized by a finite-state automaton, which operates by updating a state $s$ based on an input symbol $w$ via a transition function $\delta(s,w)$. This abstraction aligns well with our framework, where even pairs compare the last token (state) with another token in the sequence, appending the result at the end, while CoT iteratively applies this process to determine the final answer. To generalize our approach to arbitrary regular languages, the key is extending from simple token-equality comparisons to a more general transition function $\delta(\cdot,\cdot)$. This can be done by find an attention map that can approximate a general transition function $\delta$. The training dynamics analysis will leverage some key techniques that we develop here such as handling the coupling of layers and implicit bias.
>
> Q5: It would be better to phrase the method as data mixing rather than adding a regularization.
>
> A5: Thanks for the suggestion! We will make the change or add discussions.
>
> ------
> Thank you again for your insightful comments. We hope our responses addressed your concerns and would greatly appreciate your kind consideration in increasing your score.

---

> > ### Comment · Reviewer_TZ3J · 2025-04-03
> >
> > Thanks for the response. Still, I'm not fully convinced that this paper can provide useful insights into deep Transformers, and I also don't see strong evidence that the proof is flexible enough to be applied to other settings. I would like to keep my score since my two main questions remain.

---

> > > ### Author Response · Authors · 2025-04-06
> > >
> > > We thank the reviewer very much for the response and further comments. We respect your decision, and are grateful that you are keeping towards the positive opinion of our paper. We would also like to take this opportunity to further elaborate on a few points raised in your feedback.
> > >
> > > **Q1:** Useful insights into deep transformers
> > >
> > > **A1:** We first re-iterate that our analysis uncovers the distinct roles played by different layers of transformers in two phases of the training. This behavior is also observed in our new experiments on **deep** transformers on real-world data (e.g., Shakespeare text generation in Figure 4 https://anonymous.4open.science/r/icml2025-4BCC/Figures%20for%20ICML.pdf).
> > >
> > > Regarding the technical analysis of deep transformers, we would like to emphasize that theoretical understanding of their training dynamics remains a significant challenge. Current analytical tools are not yet well-equipped to handle the complexity of deep transformer architectures. In this context, our work makes two key contributions to the existing toolbox. First, we extend existing tools to analyze the joint training dynamics of attention and feed-forward layers, capturing their interplay. This goes beyond most prior studies, which only focus on one attention layer. Second, we analyze of the entire training process for **multi-step** Chain-of-Thought (CoT), capturing how CoT reasoning evolves throughout training and converges, whereas existing theoretical work captures only the first a few steps of gradient descent, without addressing the full trajectory of training.
> > >
> > > We also note that, beyond the depth of transformers, there are other important directions for advancing the theoretical understanding of transformers. In our work, a key contribution lies in demonstrating how training on a simple language task of even pairs can benefit more complex parity check task via attention mechanism. Specifically, as shown in Approach 1 (Sec 5.1), transformers trained solely on even pairs (without CoT training) can solve parity check by simply applying CoT at inference time. This stands in contrast to existing works, where CoT inference typically depends on CoT-based training. Additionally, we show that the output from even pairs can serve as an effective regularizer during training of parity check, providing a strong momentum to initialize the training.
> > >
> > > **Q2:** Whether proof is flexible enough to be applied to other settings
> > >
> > > **A2:** Our framework supports two key generalizations as we elaborate below.
> > >
> > > **Generalization to other regular languages:** Using the FSA framework describe in A4 of our first rebuttal, the transition function $\delta$ of parity and even pairs is equivalent to an XOR operator, because it examines whether two symbols are equal. However, our techniques can handle more general operations such as $\delta$ is AND, OR and even combination of these operators through CoT.
> > >
> > > For example, let us consider the case when $\delta$ is an OR function. Mathematically, if we equate 'a' with 1 and 'b' with 0, OR function $\delta$ satisfies $\delta(b,b) = b$, $\delta(a,b) = \delta(b,a)=\delta(a,a)=a$. Then, using the rationale described in line 256-274 (left column),
> > > we expect that the training to learn such a function will satisfy
> > > \\[ \langle u_t, E_1^w \rangle > 0, \langle u_t, E_\ell^w\rangle <0,\quad \langle E_1^w - E_\ell^{w'}, W_t E_L^a \rangle  > 0, \langle E_1^a - E_\ell^w, W_t E_L^b \rangle  > 0, \langle E_2^w - E_1^b, W_t E_L^b \rangle  >0 \\]
> > > In other words, if the last token and the first toke  are both 'b', then the transformer will assign more attention weights to the second token; otherwise, the transformer will assign more attention weights to the first token. Our second phase analysis can also be applied to such a setting since the max-margin problem solution will be determined by the initialization of linear and attention layer.
> > >
> > > **Generalization to other settings such as Random Initialization and Separate Key/Query/Value Matrices:** We outline more details for these extensions. Our goal is to prove similar results where the token score grows and attention score satisfies the desired property at a certain time step $t$. For example, we can show that there exists $t$ such that
> > > \\[ \langle W_o^t W_v^t, E_1^w \rangle > 0.\\]
> > > This is because sequences of length
> > > $L = 1$ (which always have positive labels) dominate the early training, and create an initial bias for the first token. Therefore, even if $W_o, W_v$ are initialized randomly, the gradient will always be negative, pushing $\langle W_o^t W_v^t, E_1^w \rangle$ to be a positive number. Then, using our argument starting in line 808, we can show that the attention layer also starts to assign more attention to the first token in positive samples.
> > >
> > > Finally, we sincerely thank the reviewer once again for the insightful discussions throughout the review process. We greatly appreciate the opportunity to engage with you on these important technical topics.

---

### Decision · Program_Chairs · 2025-05-01

**Decision:**

Accept (poster)

**Comment:**

This paper examines the training dynamics of transformers for the even pair and parity check problems. All reviewers acknowledge the theoretical contributions of this paper, including the joint training dynamic analysis and the clean theoretical setup for regular language recognition problems. The major concerns raised by the reviewers focus on the simplicity of the theoretical setting and its generalization to more complex scenarios, such as multi-layer transformers and other challenging tasks.

I agree that the contribution is somewhat limited given its idealized setting and simplified theoretical analysis. However, as a theoretical work, this paper is self-contained and explores new problems with novel techniques, which meets the acceptance criteria. Therefore, I recommend acceptance.